


# 1 What can we learn from amino acids about oceanic organic matter cycling

# 2 and degradation?

**Birgit Gaye[1], Niko Lahajnar[1], Natalie Harms[1], Sophie Anna Luise Paul[2, 3], Tim Rixen[1,4]**
**and Kay-Christian Emeis[1]**
[1]*Institute for Geology, Universität Hamburg, 20146 Hamburg, Germany*
[2]*Department of Physics and Earth Sciences, Jacobs University Bremen, 28759 Bremen,*
*Germany*
[3]*GEOMAR, Helmholtz-Zentrum für Ozeanforschung, 24148 Kiel, Germany*
[4]*Leibniz Centre for Tropical Marine Research (ZMT), 28359 Bremen, Germany*
**Correspondence:** Birgit Gaye (birgit.gaye@uni-hamburg.de)

## 14 Abstract

Amino acids (AA) mainly bound in proteins are major constituents of living biomass and non-
living organic material in the oceanic particulate and dissolved organic matter pool. Uptake and
cycling by heterotrophic organisms lead to characteristic changes in AA composition so that
AA based biogeochemical indicators are often used to elucidate processes of organic matter
cycling and degradation. We analyzed particulate AA in a large sample set collected in various
oceanic regions covering sinking and suspended particles in the water column, sediment
samples as well as dissolved AA from water column and pore water samples.  The aim of this
study was to test and improve the use of AA derived biogeochemical indicators as proxies for
organic matter sources and degradation, and to better understand particle dynamics and
interaction between the dissolved and particulate organic matter pools. A principal component
analysis (PCA) of all data delineates diverging AA compositions of sinking and suspended
particles with increasing water depth. A new sinking particle and sediment degradation
indicator (SDI) allows a fine-tuned classification of sinking particles and sediments with respect
to the intensity of degradation, which is associated with changes of bulk $\delta^{15}N$ ratios. This new
indicator furthermore is sensitive to sedimentary redox conditions and can be used to detect
past anoxic early diagenesis. A second indicator emerges from the AA spectra of suspended





particulate matter (SPM) in the epipelagic and that of the meso- and bathypelagic ocean and is
a residence time indicator (RTI). The characteristic changes in AA patterns from shallow to
deep SPM are recapitulated in the AA spectra of the dissolved organic matter (DOM) pool, so
that deep SPM is more similar to DOM than to any of the other organic matter pools. This
implies that there is equilibration between finely dispersed SPM and DOM in the deep sea,
which may be driven by microbial activity combined with annealing and fragmentation of gels.
As these processes strongly depend on physico-chemical conditions in the deep ocean, changes
in quality and degradability of DOM may strongly affect the relatively large pool of suspended
and dissolved AA in the ocean that amounts to 15 Pg amino acid carbon (AAC) and 89±29 Pg
AAC, respectively.



## 1 Introduction

Amino acids are ubiquitous in living organisms and comprise a major share of characterized organic matter in the particulate and dissolved pool in the ocean (Lee, 1988; Wakeham et al., 1984; Zhang et al., 2016; Davis et al., 2009; Lee et al., 2004). AA comprise more than 80 % of total organic carbon in fresh autochthonous plankton while allochthonous organic matter from continental runoff and atmospheric deposition has lower AA contents (Degens and Ittekkot, 1983; Degens and Ittekkot, 1985). Most of the primary productivity occurs in the sunlit surface layer of the ocean and most of the allochthonous material is also transported into surface waters. Thus, organic matter concentrations including their major biogenic constituents generally, have a surface maximum and decrease with depth (Peters et al., 2018; Gaye et al., 2013; Wakeham and Lee, 1993). The main mechanism behind this depth dependent distribution is that most of the organic matter is recycled in surface waters while only a small proportion of surface particles leaves the surface waters by gravitational settling in the form of macroaggregates or fecal pellets. Particles comprising organic matter, shells, frustules of organisms and mineral matter sink at speeds of 200 m day$^{-1}$ on average and constitute the export from the surface mixed layer or euphotic zone into the deep ocean, where part of it can ultimately reach the sediments (Alldredge and Silver, 1988; Alldredge, 1998; Pilskaln and Honjo, 1987; Fowler and Knauer, 1986; Karl et al., 1988; Rixen et al., 2019b). Sinking particles are caught by moored or floating sediment traps while SPM is sampled by filtration or ultrafiltration of water from water samplers or by pump systems (Yamaguchi and McCarthy, 2018). SPM is too small to sink and therefore – like DOM – predominantly enters deep water by subduction of surface waters (Resplandy et al., 2019; Boyd et al., 2019) and is transported passively following the route of ocean water along the ocean conveyer belt (Silver et al., 1998; Mccave, 1984). It has thus been surmised that the long residence time of SPM in the water column should result in a more degraded state compared with organic matter of sinking particles (Mccave, 1984; Degens and Ittekkot, 1984). Studies of pigments, AA and fatty acids, however, do not find such a systematic difference between the two types of particles and even indicate that SPM can be less degraded than sinking particles (Abramson et al., 2011; Rontani et al., 2011; Wakeham and Canuel, 1988). In two studies of AA composition in the Benguela Upwelling System and in the Arabian Sea it was shown that the degradation pathways of SPM and sinking particles differ as their AA compositions diverge with depth (Gaye et al., 2013; Nagel et al., 2009). These studies suggested that there is only little interaction between suspended and sinking particle pools below the euphotic zone. Due to its long residence time in the ocean, SPM appears to interact with DOM



(Gaye et al., 2013) and therefore carries different AA signatures related to genesis and history
of organic matter cycling in its specific water mass (Nagel et al., 2016). Whereas information
on the composition of sediment trap samples has been compiled in comprehensive studies
(Honjo et al., 2008; Rixen et al., 2017; Wilson et al., 2012; Rixen et al., 2019a), similar
compilations of the profuse literature on suspended matter are yet missing.
On the way to the deep sea the flux of sinking particles is reduced by disaggregation and organic
matter degradation. Suess (1980) empirically derived the first power function for organic
carbon decay based on sediment trap data. Subsequently, a large number of similar functions
where calculated for various oceanic areas based on trap experiments (Rixen et al., 2017; Rixen
et al., 2002; Armstrong et al., 2002; Martin et al., 1987). Early work on AA had produced
similar decay functions combining data from Atlantic and Pacific trap experiments (Lee and
Cronin, 1982, 1984). As AA decay faster than bulk organic carbon (Haake et al., 1993b; Haake
et al., 1992; Haake et al., 1996; Lee et al., 2004; Wakeham and Lee, 1989; Whelan and Emeis,
1992), they are often considered as "labile" constituents of bulk organic matter. This is
supposedly due to their preferential uptake as a nitrogen (N) source for further synthesis of AA
or as a source of essential AA for heterotrophs (Ittekkot and Arain, 1986; Ittekkot et al., 1986).
This has been questioned, as a large proportion of the oceanic organic N pool is comprised of
AA that are not bioavailable (Aluwihare et al., 2005). In addition to the quantification of AA
decay, degradation state of organic matter (proteins) can be assessed by characteristic changes
in AA monomer composition which, furthermore, have the potential to elucidate sources of
organic matter and degradation processes (Ittekkot et al., 1984a; Ittekkot et al., 1984b; Dauwe
and Middelburg, 1998; Dauwe et al., 1999; Jennerjahn and Ittekkot, 1997).
DOM is defined by the pore size of the filters it passes through which is 0.2-0.7 µm (Carlson
and Hansell, 2015) and thus includes some picoplankton cells and all viruses (Aristegui et al.,
2009). DOM in surface water is partly labile and can originate from the exudates and lysis of
organisms, passive diffusion, or "overflow" out of phytoplankton and bacteria; grazers can
excrete or egest DOM, it can furthermore be leached from their fecal pellets or released by
sloppy zooplankton feeding and is thus primarily released and also taken up in the surface ocean
(Carlson and Hansell, 2015). Moreover, terrestrially derived DOM is transported into surface
waters by rivers and via the atmosphere (Benner et al., 2005). Deep DOM has a different source
than simply transport of surface DOM by intermediate and deep water formation and mixing,
as deep DOM is refractory in nature and has been heterotrophically altered by cycling and
degradation processes (Yamaguchi and Mccarthy, 2018). The possible source of deep DOM



may be the release from sinking or suspended particles associated with microbial degradation
on particles and in the ambiance of particles by processes such as solubilizing organic matter
by ectohydrolase (Cho and Azam, 1988; Ciais et al., 2014; Aristegui et al., 2009). DOM can
also be released from sediment pore water into overlying waters (Lahajnar et al., 2005). Stable
isotope ratios of nitrogen ($\delta^{15}$N) in ultrafiltered DOM (UDOM) showed no systematic change
with depth and suggested a common microbial source or viral lysis (Mccarthy et al., 2007).
Studies combining nuclear magnetic resonance (NMR) spectroscopy with AA hydrolysis of
different intensity identified amide-N and amine-N as the dominant form of organic N in DOM
(Mccarthy et al., 1997; Aluwihare et al., 2005). Between 45 and 86 % of dissolved organic
nitrogen (DON) were found to be bound in proteins but only a small part could be converted
into AA by acid hydrolysis (Aluwihare et al., 2005). Differences of hydrolysis conditions can
explain the large range of AA contribution to dissolved organic carbon found in different
experiments. Mild acid hydrolysis of AA resulted in amino acid carbon contributions to total
dissolved carbon (AAC/C) between 0.4 and 4 % with a reduction from 1-4 % AAC/C % in
surface waters to 0.4-0.8 % in waters >1.000 m (Guo et al., 2018; Davis and Benner, 2005).
Moreover, this reduction was associated with a progressive AA degradation (Kim et al., 2017;
Davis and Benner, 2005). Stronger acid hydrolysis resulted in AAC/C of 5-10 % (Ittekkot,
1981; Keil and Kirchman, 1999; Mccarthy et al., 1997).
Understanding and quantifying AA degradation is required to estimate the diagenetic imprint
on $\delta^{15}$N ratios of particulate matter. N-isotope ratios are reported in ‰ using the delta notation
and the $^{15}$N/$^{14}$N of air $N_2$ as the reference standard:
$\delta^{15}N_{sample} = ((^{15}N/^{14}N)_{sample} / (^{15}N/^{14}N)_{reference\ standard} - 1) * 1000$ (1)
$\delta^{15}$N ratios track  major shifts between N pools and are commonly used to reconstruct the N
cycle from sedimentary archives (Galbraith et al., 2013). Amino acid nitrogen (AAN)
comprises 80-100 % of N in fresh organic matter and is the precursor of most of the N buried
in sediments and ultimately stored in the form of ammonium, adsorbed to clay minerals (Boyd,
2001; Waples and Sloan, 1980; Müller, 1977). Considerable AA degradation already occurs in
the water column and progresses  during organic matter burial in the sediments so that the
impact of diagenetic processes on $\delta^{15}$N has to be accounted for (Möbius et al., 2010; Möbius et
al., 2011; Niggemann et al., 2018; Carr et al., 2016). Ammonification leads to a diagenetic
increase of $\delta^{15}$N values by up to 6.5 ‰ in deep sea sediments while there is little effect during
organic matter burial in shelf and slope sediments due to the higher sedimentation rates and



sub- to anoxic diagenetic conditions (Tesdal et al., 2013; Robinson et al., 2012; Möbius, 2013;
Gaye-Haake et al., 2005). Such $\delta^{15}$N increases were shown to correlate with AA derived
degradation indicators so that the primary $\delta^{15}$N signal from the water column can be
reconstructed (Gaye-Haake et al., 2005; Gaye et al., 2009; Möbius et al., 2011).
In the following synoptic compilation of AA data, we will examine the differences in AA
spectra of a large data set that combines dissolved and particulate AA from plankton, suspended
and sinking material, and sediments from different oceanic regions, as well as from riverine to
brackish-marine conditions. Focusing on processes in the water column the data serve to (i) test
existing AA based biogeochemical indicators of organic matter sources and degradation, (ii)
better understand transformation and degradation processes of organic matter in aquatic
environments reflected by AA composition in sinking and suspended particles and total
dissolved AA (TDAA), (iii) investigate the impact of such processes on the $\delta^{15}$N values and
(iv) identify open questions which may be pursued with the help of AA analyses in the future.

### 2 Amino acid derived biogeochemical indicators of organic matter origin and degradation

Amino acid concentrations and the contribution of AA carbon (AAC) and AAN as percentages
of total organic carbon (AAC/C %) or total N (AAN/N %) are used to determine the degradation
state of organic matter in the marine realm as both decrease with increasing organic matter
degradation (Wakeham and Lee, 1993; Cowie and Hedges, 1994). AAN/N % >50 % are
characteristic of fresh organic matter in the freshwater and marine realm (Menzel et al., 2015;
Haake et al., 1992; Haake et al., 1993b). AA contribute >60 % to total organic carbon (AAC/C
%) in fresh plankton and suspended matter in surface waters whereas AAC/C % drop to values
<20 % in sinking particles and suspended matter from subsurface water (Wakeham and Lee,
1993). AAC/C % values are often below 10 % in freshwater environments and indicate the
enhanced input of land plants enriched in carbohydrates and lignin rather than enhanced organic
matter degradation (Menzel et al., 2015).
Ratios of individual AA such as Asp/β-alanine (β-Ala) and Glu/γ-aminobutyric acid (γ-Aba)
(Cowie and Hedges, 1994; Ittekkot et al., 1984a; Lee and Cronin, 1984), or the Reactivity Index
(RI) (Jennerjahn and Ittekkot, 1997) and the Degradation Index (DI) (Dauwe et al., 1999;
Dauwe and Middelburg, 1998) have often been used to scale organic matter degradation
(Niggemann et al., 2018; Unger et al., 2005; Ingalls et al., 2006; Ingalls et al., 2004; Pantoja et





al., 2004; Möbius et al., 2010). After acid hydrolysis we can neither distinguish between Asp
and asparagine (Asn) nor between Glu and glutamine (Gln) so that our Asp measurements
comprise Asp+Asn and our Glu comprise Glu+Gln. Asn, Gln and Glu are the primary products
of N assimilation and all other AA are synthesized from them (Loick-Wilde et al., 2018;
Riccardi et al., 1989; Hildebrandt et al., 2015). Asp and Glu are the major AA in bacteria,
vascular plant tissue, phytoplankton, zooplankton and fungi (Cowie and Hedges, 1992). High
relative contents of Asp and Glu, therefore, indicate fresh organic matter and the ratios of
Asp/β-Ala and Glu/γ-Aba are high in fresh organic matter as β-Ala and γ-Aba are degradation
products of Asp and Glu, respectively (Lee and Cronin, 1984). β-Ala and γ-Aba also become
relatively enriched during organic matter degradation as these non-protein AA are not taken up
by heterotrophic organisms (Ittekkot et al., 1984b). Although Asp/β-Ala and Glu/γ-Aba are
often used in concert to study degradation processes, they are likely to deviate as Glu
accumulates in plant material while Asp accumulates in degraded sediments (Möbius et al.,

183    2011).

The relative accumulation of the non-protein AA is also expressed by the RI which is the ratio
of the very labile aromatic AA Tyr and Phe and the non-protein AA β-Ala and γ-Aba. The RI
is a useful indicator of organic matter quality and is, generally, between 0 (very degraded) and
15 (very fresh) (Jennerjahn and Ittekkot, 1997). It is applicable not only in studies of sinking
and suspended matter in marine and brackish environments (Unger et al., 2005; Gaye et al.,
2007) but also as a proxy for degradation state in the sediment column (Möbius et al., 2011).
The enrichment of Asp and Glu in sediments is related to their enrichment in carbonate shells
(Ittekkot et al., 1984a) and to adsorption of primarily acidic AA onto carbonate minerals (King
and Hare, 1972), whereas basic AA primarily adsorb onto silicate minerals (Hedges and Hare,
1987; Keil et al., 1994; King, 1975).
The DI, the integral of 14 protein AA, assesses the diagenetic alteration of a sample by
comparing it to a set of 28 sediment samples of different degradation states and environments.
Molar percentages of individual AA are standardised by the mean and standard deviations of
the 28-sample data set. The DI then integrates the result of this standardized values weighed by
the factor coefficients for the first axis of the PCA of Dauwe et al. (1999) according to the
formula:
$$DI = \sum_i \left[ \frac{var_i - AVGvar_i}{STDvar_i} \right] \cdot fac.coef._i$$

(2)





where $var_i$ is the original mole percentage of each $AA_i$, $AVGvar_i$ and $STDvar_i$ are the mean and
standard deviations, respectively, and $fac.coef._i$ is the factor coefficient of the first axis of the
PCA of Dauwe et al. (1999). The DI thus represents the cumulative deviation of AA with
respect to an assumed average molar composition. The DI ranges approximately from -2 to +3
where negative values indicate more and positive values less degradation than the average.
These biogeochemical indicators of organic matter quality were essentially developed for
marine sinking particles and sediments. They are of limited use in other sample sets and
materials, such as marine SPM and samples from fresh and brackish waters, so that individual
and adapted indices were developed to differentiate states of degradation (Abramson et al.,
2011; Gaye et al., 2007; Goutx et al., 2007; Menzel et al., 2013; Sheridan et al., 2002). Such
indices are usually derived from a principal component analyses (PCA) of Mol% AA. Menzel
et al. (2015) used data from a suite of lake samples from different climate regimes in India for
a PCA and suggested to use the factor coefficients of the AA to calculate a lake sample
degradation index (LI). Gaye et al. (2007) calculated a new degradation index for sediment
traps from the Kara Sea whereas Unger et al. (2005) used the established DI and RI to classify
sediment and suspended water samples from the river endmembers Ob and Yenisei into the
brackish-marine Kara Sea. The DI was even applied to trace dissolved AA degradation (Davis
and Benner, 2005; Guo et al., 2018), although the DI is originally based on marine sediments
only (Dauwe et al., 1999). An indicator of oxic vs. anoxic organic matter degradation in the
water column and in sediments was proposed by Menzel et al. (2015) for lake samples. Based
on work by Cowie et al. (1995) on marine sediments the ox/anox indicator is the quotient of
AA preserved under oxic diagenetic conditions to those preserved in anoxic water or sediments
and is thus higher in oxic than in anoxic sediments:

$$ox/anox = \frac{Asp + Glu + \beta\text{-}Ala + \gamma\text{-}Aba + Lys}{Ser + Met + Ile + Leu + Tyr + Phe}$$

224                                                                                              (3)

with Lys being lysine and Ser being serine. Ox/anox ratios <1.0 indicate anoxic and ratios >1.5
oxic diagenesis (Menzel et al., 2015).
The stability of AA vs. hexosamines (HA) has been discussed since the early research on AA
and HA in natural material. Fresh plankton was observed to have AA/HA ratios of 13-25
(Degens and Mopper, 1975) which is a mixed signal of phytoplankton with an AA/HA ratio of
>80 and zooplankton with a ratio of ~9 due to chitinaceous skeletons of many zooplankters
(Mayzaud and Martin, 1975). Low AA/HA are also observed in cell walls of fungi and bacteria.





As the building blocks of chitin, HA were assumed to be more resistant to degradation than
bulk AA (Muller et al., 1986). This is, however, challenged by studies of enzyme activities
which were observed to respond to substrate availability so that the activity of chitobiase and
chitinase as high as that of glucosidase (Boetius et al., 2000a; Boetius and Lochte, 1994; Boetius
et al., 2000b; Smith et al., 1992) suggesting intense degradation also of chitin. Glucosamine
(Gluam) is the main constituent of chitin and while Galactosamine (Galam) is relatively
enriched in bacterial cell walls (Walla et al., 1984; Kandler, 1979). The Gluam/Galam ratio has,
therefore, been used to distinguish bacterial material from zooplankton rich material (Haake et
al., 1993b; Benner and Kaiser, 2003; Niggemann and Schubert, 2006). Gluam/Galam ratios >
4 were found in sinking particles (Haake et al., 1993b; Haake et al., 1992; Lahajnar et al., 2007),
ratios of < 3 usually indicate relatively high contribution of microbial OM and values between
1 and 2 are characteristic of sediments and indicate an enrichment of microbial biomass (Benner
and Kaiser, 2003).

**3. Materials and Methods**
**3.1 Sampling**
A total of 1425 samples were taken in different oceanic areas and water depths and include 218
sediment trap samples, 489 sediment samples, 608 SPM samples and 110 water and pore water
samples (Fig. 1). Five additional plankton samples were taken in the Arabian Sea and from the
Namibian upwelling area by plankton tows between 0-100 m and between 100-700 m water
depths. Sea water was filtered through Whatman GF/F filters and dried at 40°C in order to
obtain SPM samples. An aliquot of the filtrate was stored deep frozen for AA analyses.
Sediment trap samples were wet sieved on board and comprise the <1mm fraction filtered with
polycarbonate nuclepore filters of 0.45 µm pore size and dried at 40°C. Sediment samples from
multicores, box grabs, box cores, or gravity cores were taken by spatula or syringes from cold
stored cores and were freeze dried before analyses. Surface samples represent either the upper
0.5 cm or 1 cm of a sediment core. Pore-water samples were taken by rhizons and stored frozen
before analyses (see methods in Paul et al., 2018). 18 water samples taken off Namibia were
separated into two size classes by ultrafiltration (Brockmeyer and Spitzy, 2013). The size
classes 50 kDa-0.7 µm and 1 kDa-0.7 µm were used for AA analyses.





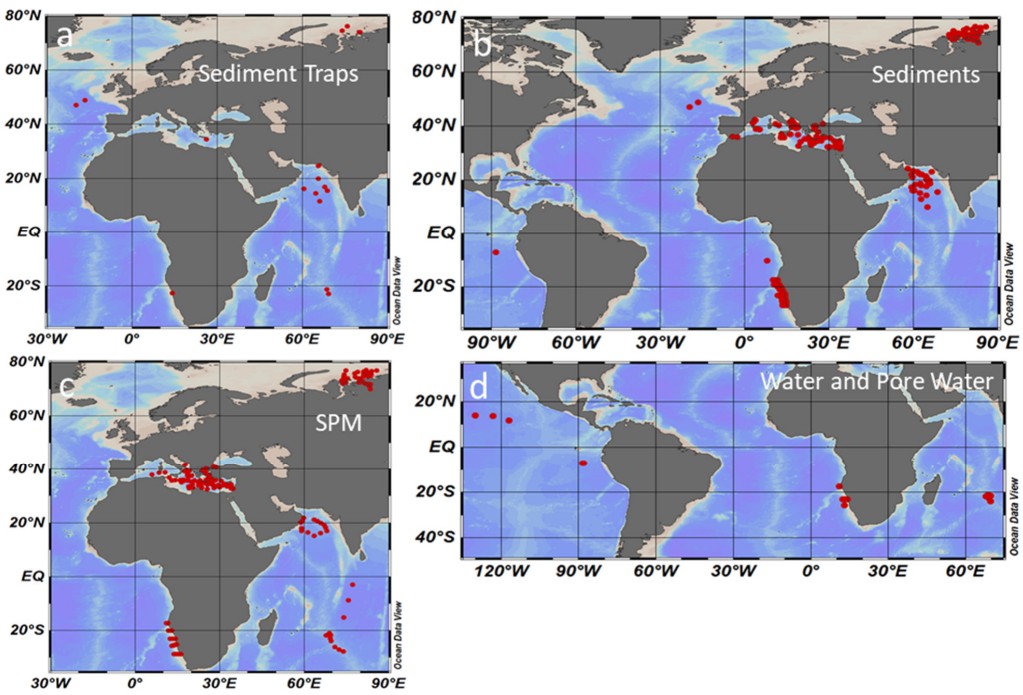


Figure 1: Stations of sediment trap deployments (a), sediments (b), SPM (c) and water and pore
sampling (d).

**3.2 Analytical methods**
Total carbon and N were measured with a Carlo Erba Nitrogen Analyser 1500 (Milan, Italy) or
a EURO EA3000 elemental analyzer. Particulate organic carbon (POC) was measured after
treatment of weighed samples with 1N HCl to remove carbonate. The precision of this method
is 0.05% for carbon and 0.005% for N. Carbonate carbon was calculated by subtracting organic
carbon from total carbon. Ratios of $^{15}N/^{14}N$ of particulate N were determined using a Thermo
Finnigan MAT 252 isotope ratio mass spectrometer connected with a ConFlo-III interface after
high-temperature flash combustion in a Thermo Finnigan Flash EA 1112 at 1050°C. Part of the
samples were measured with an Elementar IsoPrime 100 isotope ratio mass spectrometer after
high temperature combustion in an Elementar CHNOS Vario isotope elemental analyser at 950
°C. Pure tank $N_2$ calibrated against the reference standards IAEA-N1 (ammonium sulfate,
$\delta^{15}N = + 0.4$ ‰ versus air $N_2$) and IAEAN2 (ammonium sulfate, $d^{15}N = + 20.3$ ‰) of the
International Atomic Energy Agency was used as a working standard. Duplicate measurements



of samples differ by less than 0.15 ‰. The laboratory's long-term standard deviation for IAEA-
N1 standard is 0.09 ‰.
Dissolved organic carbon (DOC) concentrations [mg/L] were determined via a high
temperature combustion method (POC-V$_{CSH}$ Analyzer, Shimadzu). Sampled water was filtered
through pre-combusted Whatman GFF filters and inorganic carbon was removed by 2 M HCl
prior to injection into the combustion tube where organic carbon is oxidized to $CO_2$ at 680 °C
with a platinum catalyst. A 5-point calibration from 0.5 to 5 mg DOC/L was used. The error of
measurement is less than 2 % (Brockmeyer and Spitzy, 2013).
Total dissolved AA, particulate AA and hexosamines (HA) of selected samples were analyzed
with a Biochrom 30 Amino Acid Analyzer. Acid hydrolysis with 6N HCl for 22 h at 110°C
under a pure argon atmosphere was carried out on ca. 3 ml of filtrate of water samples, on 1-2
mg of suspended matter collected on GF/F filters, on 1-2 mg of sediment trap samples, or on 1-
50 mg of freeze dried surface sediments. A particle free aliquot was evaporated three times to
dryness in order to remove the unreacted HCl; the residue was taken up in an acidic buffer (pH
2.2). After injection and subsequent separation with a cation exchange resin, the individual AA
monomers were post-column derivatized with o-phthaldialdehyde in the presence of 2-
mercaptoethanol and detected with a Merck Hitachi L-2480 fluorescence detector. Duplicate
analysis of a standard solution according to this method results in a relative error of 0.1 to 1.3%
for the concentrations of individual AA monomers and 0.2 to 3.0% for individual AA
monomers of water or particulate matter samples. Due to acid hydrolysis, Asp and Asn are both
measured as Asp and Glu and Gln are both measured as Glu. The other AA measured are Thr,
Ser, Gly, Ala, valine (Val), Met, Ile, Leu, Tyr, Phe, β-Ala, γ-Aba, histidine (His), ornithine
(Orn), Lys and arginine (Arg). The measured HA are Gluam and Galam.

### 3.3 Statistical analyses

To investigate the differences of AA composition and to recognize the interaction and pathways
of degradation between the different pools we carried out a PCA of AA monomer contributions
in Mol %. Met was excluded as it is below detection limit in many samples. The PCA was
carried out using the program SPSS Statistics 22. PCAs have often been used to analyze large
databases (Xue et al., 2011) in order to trace organic matter degradation, group and categorize
samples and develop indices such as the DI of Dauwe et al. (1999) using summary statistics





(see equation 1). A PCA is an orthogonal transformation of a set of variables into a new set of
uncorrelated variables called principal components. New axes are chosen in order to explain as
much as possible of the variance within the data set on a few main axes of highest correlation.
The first component explains most of the variance within the data set, consecutively followed
by the remaining components in the order of their decreasing capacity to explain the variance
within the data set. The selection of the most relevant components can be done by selecting
those with eigenvalues (the variances of the principal components) >1. Alternatively, the kink
method can be applied selecting those components from a plot of eigenvalues (scree plot),
which describe a steep slope of declining variance followed by a "kink" after which the
principal components add only small amounts to the variance. The factor loadings of the
variables (in this case the individual amino acids) are their projections on the new axis. The
factor score of each data set from a sampling location is obtained by multiplying the
standardized data with the factor loadings (also called factor coefficients). A high (low) factor
score shows that a sample has high (low) concentrations of the variables with high factor
loadings. A plot of factor loadings of the variables compared with a plot of the factor scores of
samples helps to visualize the relation of the samples to the variables and thus to identify the
processes behind the results of the PCA.

**4 Results**
**4.1 Organic carbon, nitrogen and amino acid content**
The POC (N) content is 35.9 % (5.9%) in plankton and 1.65-46.4 % (0.21-10.14 %) in sediment
trap samples. In sediments, POC (N) contents drop to 0.10-13.5 % (0.02-1.72 %). SPM has
POC (N) contents of 0.94-45.4 % (0.09-12.08 %). DOC content in water samples is between
0.5-1.1 mg L$^{-1}$ and DOC in pore water samples is between 3.9-29 mg L$^{-1}$.



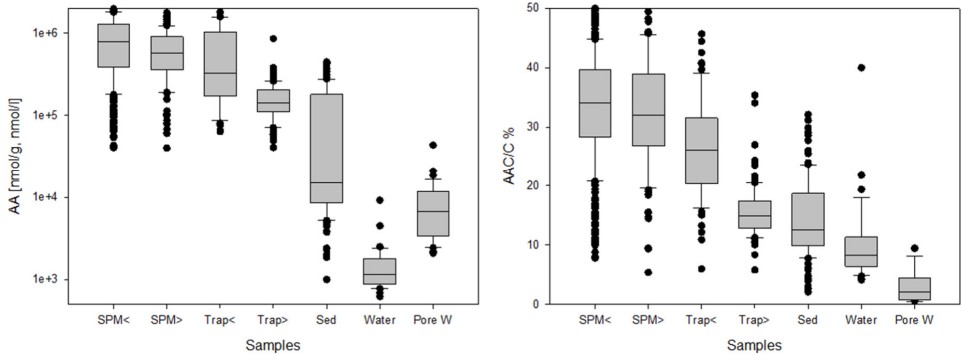

Figure 2: Box and Whisker plots of AA content in nmol g$^{-1}$ or nmol L$^{-1}$ (a) and of AAC/C %
(b) in SPM from water depths <200m (SPM<) and >200m (SPM>), in sediment traps at water
depth <200m (Trap<) and >200 m (Trap>), in sediments (Sed), in water samples (Water) and
in pore water samples (Pore W). Boxes comprise the upper and lower quartile and lines indicate
median; whiskers delineate the 10 and 90 percentile; outliers are marked by dots.

AA contents are grouped into SPM and trap samples taken at water depths <200m (shallow)
and >200m (deep) (Fig. 2, Table 1). AA contents are highest in SPM samples and shallow
sediment traps (<200m water depth) with values between 40-4307 µmol g$^{-1}$ (Fig. 2) and
averages of 662-908 µmol g$^{-1}$ (Table 1). AA contents are lower in traps from water depth >200m
with an average of 164 µmol g$^{-1}$. Sediments have lowest AA contents of all particulate matter
samples with an average of 50 µmol g$^{-1}$ (Table 1). TDAA concentrations are between 0.6-44
µmol L$^{-1}$ and AA contents are lower in water than in pore water samples with averages of 3.2
and 8.8 µmol L$^{-1}$, respectively.

The AAC/C is between 5.4-66 % in SPM and traps samples and the AAN/N (not shown) is
between 3.7-100 %. The overall pattern found for AAC/C (Fig. 2b) is similar to the pattern of
AA contents (Fig. 2a) but there is more overlap of AAC/C between the different groups.
Sediments have AAC/C between 2.7-50 % and AAN/N between 3-78 % (not shown). The
contribution of AAC to DOC (AAC/C) in water samples is between 4-40 % and in pore water
samples between 0.5-9 %.

AA contents of sinking and suspended particles decrease with water depth and the most
significant decrease occurs in the upper ocean (Fig. 3a). The decay constant of AA of sinking
particles is twice as high as the decay constant of AA of SPM (Fig. 3 a, b). Kara Sea samples
were excluded from these calculations as their AA contents are low due to the strong dilution
by material from rivers and resuspended sediments in this near-shore environment. It is also

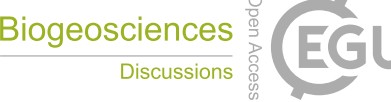

notable that AAC/C and AAN/N (not shown) significantly decrease between shallow and deep

traps and from deep traps to sediments while AAC/C of SPM show little decrease between

shallow and deep samples (Fig. 2b).

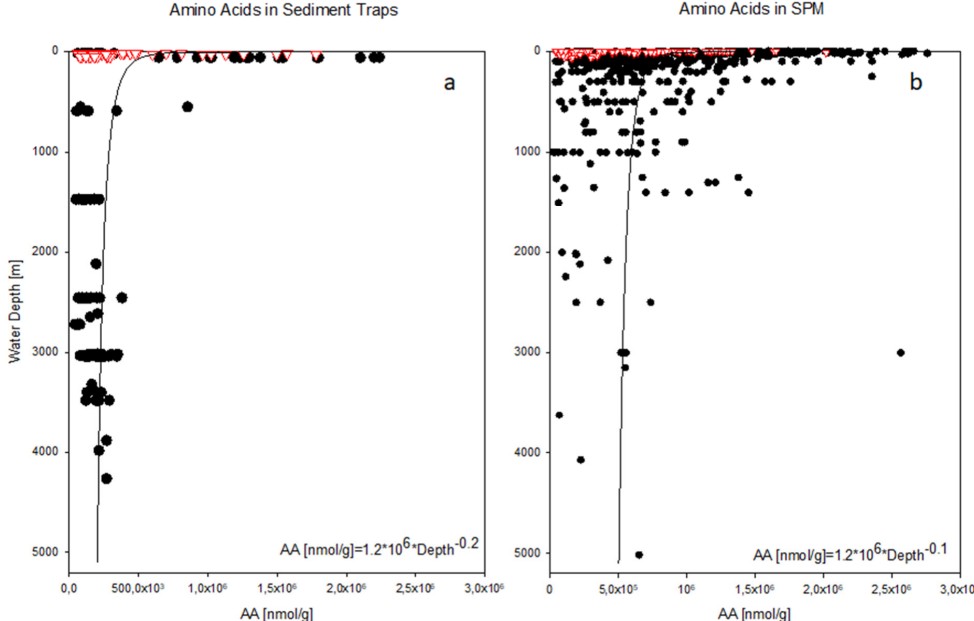

Figure 3: AA contents in nmol g$^{-1}$ in sediment traps (a) and SPM (b). Red triangles mark
samples from the Kara Sea, black dots are samples from the other trap and SPM locations shown
in Figure 1. The decay functions are calculated from samples excluding Kara Sea samples.

## 4.2 Amino acid composition

Dominant AA in plankton samples are Glu, Gly, Val and Asp. Sinking particles and sediments

have increasing Mol% of Gly, Asp, β-Ala, γ-Aba and Orn contents while the Mol% of Glu,

Ala, Val, Met, Ile, Leu, Tyr and Phe decrease (Fig. 4a). The AA enriched from plankton via

SPM to water and pore water samples are Ser, Gly, β-Ala, γ-Aba, Orn and His while all other

AA decrease (Fig. 4b). Amino acid spectra in water and pore water samples are very similar

(Fig. 4c).



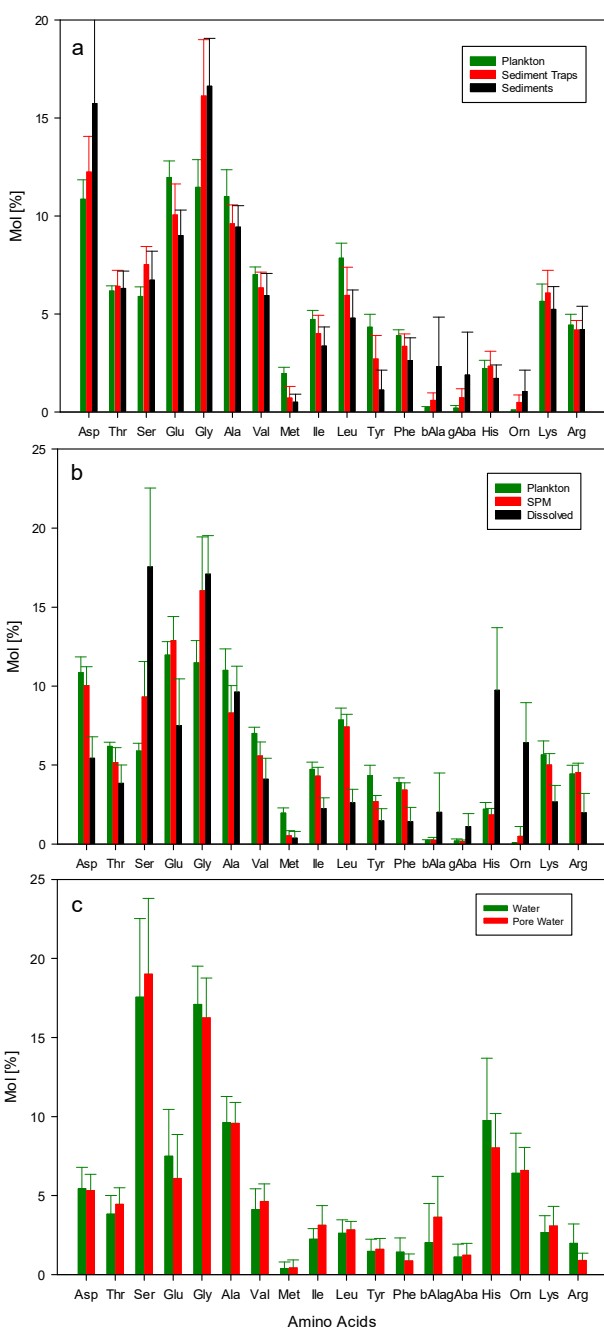

376

Figure 4: Average concentrations of individual AA (Mol%) and 1 σ standard deviation (vertical
bars) in plankton, sediment trap and sediment samples (a), in plankton, SPM and water samples
(b) and in water samples (green) and in pore water (red) (c).



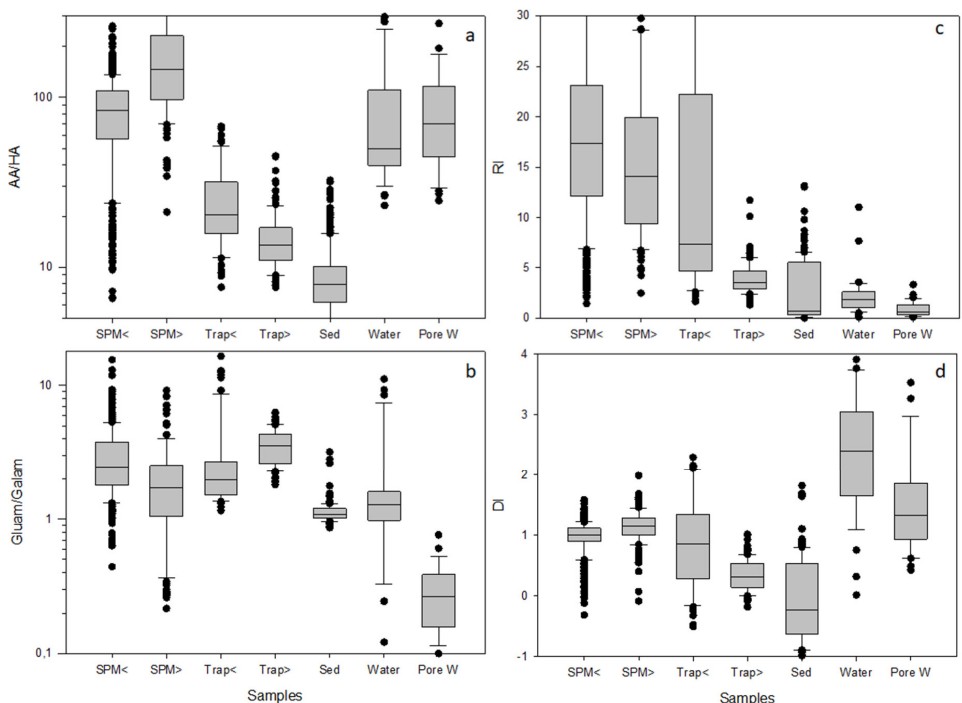

Figure 5: Box and Whisker plot of AA/HA ratios (a) and Gluam/Galam ratios (b), RI (c) and DI (d) in SPM from water depths <200m (SPM<) and >200m (SPM>), in sediment traps at water depth <200m (Trap<) and >200 m (Trap>), in sediments (Sed), in water samples (Water) and in pore water samples (Pore W). Outliers are marked by dots. Note logarithmic scales of AA/HA (a) and Gluam/Galam (b).

The biogeochemical indicator AA/HA is higher in SPM than in sediment trap samples with averages of 103 in shallow and 105 in deep SPM while Gluam/Galam ratios slightly decrease with depth (Fig. 5a, b; Table 1). The AA/HA decrease from shallow via deep traps to sediments with averages of 25.4, 14.9 and 9.1, respectively, while the Gluam/Galam is in a similar range in deep traps compared to shallow traps but is lower in sediments than in traps (Fig. 5a, b; Table 1). The RI (Fig. 5c; Table 1) shows the same pattern as the ratios of Asp/β-Ala and the Glu/γ-Aba (Table 1) has high values in SPM and shallow traps (averages of 15.1-18.8) and no significant trend between shallow and deep SPM samples. The RI decreases from shallow to deep traps and further to the sediments. Water samples have similar values as sediment samples with average RI of 1.8 and 1.6, respectively, and pore waters have an even lower average RI of 0.9. Similar to the RI the DI is not significantly different in shallow and deep SPM samples



(Fig. 5d; Table 1) while it decreases from shallow sediment traps via deep traps to sediments.
In contrast to the RI where water samples have lowest values the highest DI values are found
in water and pore water samples.
In summary, common biogeochemical indicators of organic matter degradation (RI, Asp/β-Ala,
Glu/γ-Aba) drop and thus imply increasing degradation between shallow and deep sediment
traps and between deep traps and sediments, while these indicators reveal little or no
degradation with depth in SPM (Fig. 5; Table 1). The enhanced DI values, furthermore, imply
that water and SPM samples are less degraded than deep trap and sediment samples and that
dissolved AA in water samples are least degraded.

Table 1: Mean values and standard deviation (Stdev.) of POC [%], DOC [mg/L], amino acid
(AA) content [μmol/g or μmol/L], AAC/C%, AAN/N%, ratios of Asp/β-Ala, Glu/γ-Aba,
AA/HA and Gluam/Galam, the RI, DI, SDI*, RTI* and ox/anox ratio summarized in traps at
<200m and >200m water depth, sediments, SPM <200m and >200m water depth, water samples
and pore water samples. *definition of these indicators in part 5.2 below.

| | | Trap <200m | Trap >200m | Sediment | SPM <200m | SPM >200m | Water | Pore Water |
|---|---|---|---|---|---|---|---|---|
| POC; DOC [%; mg/L] | Mean | 13,6 | 5,3 | 1,8 | 14,9 | 10,6 | 0,8 | 13,0 |
| | Stdev. | ±11.4 | ±1.9 | ±2.2 | ±8.3 | ±4.8 | ±0.2 | ±8.3 |
| Amino Acids | Mean | 631,3 | 164,1 | 49,8 | 907,7 | 661,9 | 3,2 | 8,8 |
| [μmol/g; μmol/L] | Stdev. | ±602.0 | ±93.8 | ±82.3 | ±637.6 | ±434.7 | ±3.2 | ±7.9 |
| AAC/C [%] | Mean | 26,5 | 15,5 | 10,8 | 33,9 | 32,7 | 10,1 | 2,8 |
| | Stdev. | ±8.6 | ±4.2 | ±5.9 | ±12.6 | ±10.5 | ±6.5 | ±2.6 |
| AAN/N [%] | Mean | 57,0 | 38,3 | 24,2 | 65,6 | 61,6 | | |
| | Stdev. | ±14.8 | ±8.8 | ±12.8 | ±18.4 | ±26.4 | | |
| Asp/β-Ala | Mean | 71,8 | 19,2 | 10,5 | 57,9 | 47,3 | 10,2 | 2,6 |
| | Stdev. | ±63.4 | ±19.5 | ±6.6 | ±64.7 | ±44.9 | ±14.7 | ±2.4 |
| Glu/γ-Aba | Mean | 45,7 | 12,9 | 8,5 | 103,6 | 105,3 | 8,5 | 8,8 |
| | Stdev. | ±38.8 | ±7.2 | ±6.9 | ±122.7 | ±69.6 | ±7.0 | ±10.2 |
| AA/HA | Mean | 25,4 | 14,9 | 9,1 | 84,6 | 204,6 | 80,2 | 106,8 |
| | Stdev. | ±14.7 | ±6.6 | ±4.6 | ±42.7 | ±179.1 | ±65.2 | ±142.7 |
| Gluam/Galam | Mean | 3,2 | 3,6 | 1,2 | 3,0 | 2,1 | 1,6 | 0,3 |
| | Stdev. | ±3.3 | ±1.1 | ±0.3 | ±1.8 | ±1.7 | ±2.4 | ±0.2 |
| RI | Mean | 15,1 | 3,9 | 1,8 | 18,8 | 15,7 | 1,6 | 0,9 |
| | Stdev. | ±16.4 | ±1.5 | ±2.2 | ±10.6 | ±8.3 | ±1.8 | ±0.7 |
| DI | Mean | 0,9 | 0,3 | -0,5 | 1.0 | 1,1 | 2,1 | 1,5 |
| | Stdev. | ±0.8 | ±0.3 | ±0.8 | ±0.1 | ±0.3 | ±1.1 | ±0.9 |
| SDI | Mean | 1,1 | 0,0 | -0,9 | 0,8 | 0,8 | -0,7 | -0,8 |
| | Stdev. | ±0.2 | ±0.2 | ±1.0 | ±0.2 | ±0.3 | ±0.4 | ±0.3 |
| RTI | Mean | 0,7 | 0,1 | 0,6 | 0,2 | -1,0 | -2,8 | -2,7 |
| | Stdev. | ±0.3 | ±0.3 | ±0.4 | ±0.5 | ±0.4 | ±0.6 | ±0.4 |
| ox/anox | Mean | 1,2 | 1,3 | 2,2 | 1,1 | 1,0 | 0,8 | 0,7 |
| | Stdev. | ±0.3 | ±0.2 | ±1.1 | ±0.1 | ±0.1 | ±0.3 | ±0.3 |








## 5 Discussion

### 5.1 Changes during organic matter degradation

Our summary of AA data from various locations in the world ocean corroborates earlier findings that the AA spectra of plankton and sinking particles are similar in surface waters while degradation of organic matter by zooplankton and microbes imparts characteristic changes to AA spectra with increasing water depth (Lee, 1988). The AA spectra track the successive degradation of organic matter during sedimentation from the plankton source via sinking particles, their incorporation into sediments and their further degradation after burial. The most characteristic changes along this sedimentation pathway are the relative enrichments (in Mol%) of Gly, Asp and the non-protein AA β-Ala, γ-Aba and Orn and the relative decrease of AA produced by fresh plankton such as Glu, Ala, Val, Met, Ile, Leu, Tyr and Phe (Fig. 4a). These changes are depicted by the common biogeochemical indicators: the ratios of proteinaceous AA vs. non-protein AA (RI and Glu/γ-Aba) decrease along this pathway. Asp/β-Ala ratios also decrease because β-Ala becomes relatively more enriched than Asp. The DI, originally derived from sediment samples of different degradation states (Dauwe et al., 1999; Dauwe and Middelburg, 1998), decreases from positive values in fresh plankton and most sinking particles to negative values in sediments as it integrates the products of Asp and Gly multiplied with negative factors, and the products of Glu, Met, Ile, Leu, Tyr and Phe multiplied with positive factors (Dauwe et al., 1999).

In contrast, the AA in SPM evolve along a different path than the sedimentation pathway (Gaye et al., 2013). Shares of Ser, Gly, ß-Ala, γ-Aba, His and Orn increasewith water depth, whereas almost all other AA become relatively depleted (Fig. 4b). These trends in AA spectra of SPM are also seen in the DOM phase of sea water and lead to maxima of Ser, Gly and His (Figs. 4b, c). The striking difference in AA distribution of SPM (Fig. 4) on the one hand and sinking particles and sediments on the other hand suggest that there is little exchange between the two types of particles in the ocean. Sinking particles build up sediments and the degradation pathways evident in the water column - namely the accumulation of degradation products and acidic AA often absorbed to carbonates - continue in the sediments. SPM, however, follows a different pathway that is not captured by the common AA based biogeochemical indicators of organic matter degradation (Table 1; Fig. 5) so that novel biogeochemical indicators are required to characterize their AA changes.



## 5.2 Results of a PCA: two new biogeochemical indicators

A PCA of individual AA (Mol %, Fig. 6a) of all samples compiled in this study results in two factors which explain 59 % of the total variance within the data set. The first factor delineates the well-known changes along the degradation pathway from plankton via sinking particles to sediments. Phe, Ile, Leu, Glu and Tyr (enriched in fresh plankton) have the highest F1 loadings while Asp, ß-Ala and γ-Aba (accumulating during degradation) have the lowest negative F1 loadings. Highest F2 loadings are found for Asp, Thr, Lys and Val while Gly, Orn, His and Ser have the most negative F2 loadings. Factor scores of the individual samples (Fig. 6b) plot in a triangular shape with plankton and fresh organic matter from surface waters at the apex with highest F1 and F2 scores. The diverging sides of the triangle mark particles and sediments decreasing in F1 scores on one side and SPM with decreasing F2 scores on the other side (Fig. 6b). Similar trends were observed in earlier studies based on local data sets (Nagel et al., 2016; Gaye et al., 2013). That samples from greatly different environments reveal the same divergence between sinking particles and SPM with only little overlap (Fig. 4) suggest a general mechanism operating globally. Most of the overlap encompasses SPM samples from the Kara Sea which were sampled at water depths below 100 m. The Kara Sea is characterized by sediment resuspension related to strong riverine input in combination with sea ice dynamics so that many of the Kara Sea SPM samples are mixed with resuspended sediments (Gaye et al., 2007; Unger et al., 2005). TDAA analyzed in water and pore water form a cluster with significantly different AA composition from particulate matter, but instead recapitulating the enrichments of Orn, His, Ser and Gly observed in SPM (Fig. 6b).

The precise separation of the degradation pathway of sinking particles and sediments from SPM and DOM by the PCA suggests that we can use the first factor (F1) to calculate a new sinking particle and sediment degradation index (SDI)

$$SDI = \sum_i \left( \frac{var_i - AVG.var_i}{STD.var_i} \right) \times Loadings.F1_i \qquad (4)$$

where $var_i$ is the original mole percentage of each $AA_i$, $AVGvar_i$ and $STDvar_i$ are the mean and standard deviations, respectively, and $Loading.F1_i$ is the factor loading of the first axis (F1) of the PCA of the individual amino acid$_i$ shown in Table 2. The second factor (F2) - normalized in the same way with the averages and standard deviations of the same PCA - can be used as an indicator of changes in the AA composition of SPM depending on its residence time in the ocean (see discussion below) and is therefore named residence time index (RTI)



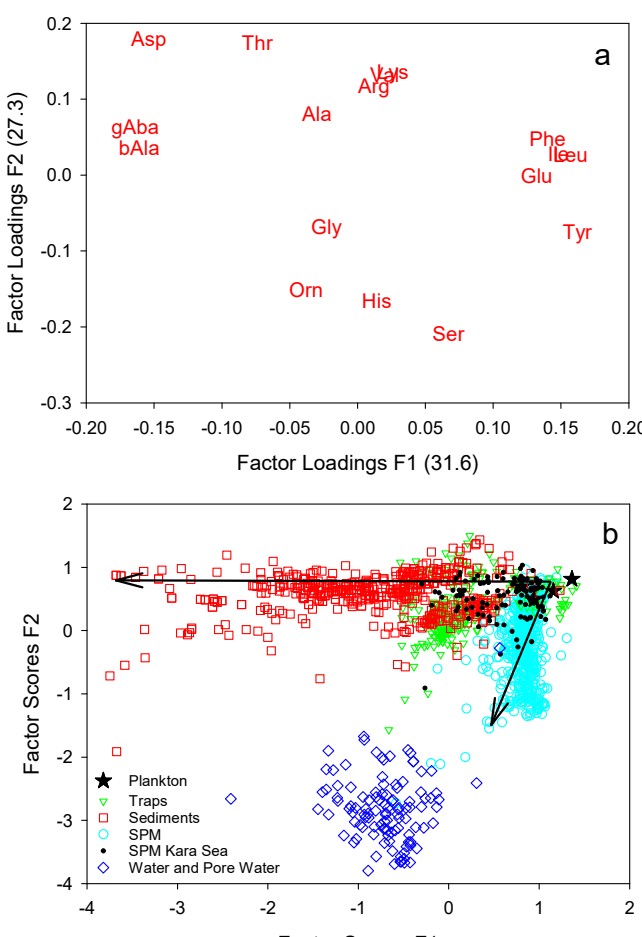


Figure 6: Results of a PCA of AA (Mol%) of all samples of this study with factor loadings of
amino acids for the first and second factor (a) and factor scores of samples (b). Arrows indicate
progressive deviation in composition from the plankton source, essentially with increasing
water and sediment depths.

$$RTI = \sum_i \left( \frac{var_i - AVG.var_i}{STD.var_i} \right) \times Loadings.F2_i \qquad (5)$$
and is calculated in the same way as the SDI but the factor loadings of the second axis (F2) of
the PCA of the individual amino acid$_i$ (Table 2) is inserted for the term *Loadings.F2$_i$*. Most of
the F1 loadings resemble those of the DI of Dauwe et al. (1999) (Table 2) and the SDI and DI
thus are significantly correlated.



A close look at the results in Fig. 6 (see also Figure S1) shows that the new indicator SDI allows
a separation of trap samples from <200 m water depth from those from greater depths. At an
SDI value of 0.5, lower values indicate stronger degradation in the deep samples. Likewise,
SPM from >200 m depths has lower RTI than most of the samples from shallower depths. Deep
trap samples and deep SPM samples form two clearly separated clusters with different SDI and
RTI (see Figure S1 for further details).
Table 2: Factor loadings of F1 and F2 for calculating the SDI and RTI, respectively, average
[Mol%] and standard deviations (Std. Dev.) of AA of samples used for the PCA shown in Figure
5 in comparison with the factor loadings (named factor coefficients) of the DI published by
Dauwe et al (1999) and their averages [Mol%] and standard deviations used for the DI based
on 28 sediment samples.

| Amino Acid | Loadings F1 SDI | Average [Mol%] | Std. Dev. | Loadings F2 RTI | DI | DI Average [Mol%] | DI Std. Dev. |
|---|---|---|---|---|---|---|---|
| Ser | 0.067 | 8.7 | 3.6 | -0.210 | 0.015 | 7.2 | 1.9 |
| His | 0.014 | 2.4 | 2.3 | -0.166 | 0.158 | 1.0 | 0.8 |
| Orn | -0.038 | 1.2 | 1.9 | -0.152 | - | - | - |
| Tyr | 0.162 | 1.9 | 1.1 | -0.075 | 0.178 | 2.1 | 1.2 |
| Gly | -0.023 | 16.2 | 2.8 | -0.068 | -0.099 | 17.6 | 3.8 |
| Glu | 0.132 | 10.5 | 2.5 | -0.001 | 0.065 | 10.0 | 2.3 |
| Leu | 0.157 | 5.9 | 1.9 | 0.027 | 0.169 | 6.6 | 1.5 |
| Ile | 0.148 | 3.8 | 1.0 | 0.028 | 0.139 | 4.5 | 0.8 |
| β–Ala | -0.161 | 1.3 | 2.0 | 0.036 | - | - | - |
| Phe | 0.140 | 2.9 | 1.0 | 0.047 | 0.134 | 3.2 | 1.2 |
| γ–Aba | -0.164 | 1.0 | 1.6 | 0.064 | - | - | - |
| Ala | -0.030 | 9.1 | 1.5 | 0.080 | -0.043 | 11.8 | 0.8 |
| Arg | 0.012 | 4.2 | 1.1 | 0.117 | -0.115 | 6.1 | 2.3 |
| Val | 0.020 | 5.7 | 1.1 | 0.132 | -0.044 | 7.6 | 1.1 |
| Lys | 0.026 | 5.1 | 1.2 | 0.135 | - | - | - |
| Thr | -0.074 | 5.7 | 1.2 | 0.174 | -0.129 | 7.1 | 1.5 |
| Asp | -0.154 | 12.2 | 4.3 | 0.179 | -0.102 | 13.4 | 2.7 |



### 5.2.1 The SDI as an indicator of degradation and oxic vs. anoxic diagenetic conditions of sinking particles and sediments

In order to test the performance of our new degradation indices and to find out if individual AA
can be used in place of the SDI and RTI, we separated SPM samples from sinking particles and
sediments and correlated the common biogeochemical indicators and individual AA (Mol %)
of SPM with the RTI of individual samples while we correlated the same variables of sinking
particles and sediments with the SDI (Table 3). We assume that correlations with Pearson



correlation coefficients R>0.50 can be considered as "strong correlations" (Cohen, 1988). The
SDI correlates moderately to strongly with the common degradation indicators and the best
positive correlation is found between SDI and the DI (Table 3). The strong correlation among
the degradation indicators with POC contents indicates that this common and often measured
variable is a good indicator of relative organic matter quality in sinking particles and sediments
and all other degradation indices (except the SDI and DI) do not perform better than POC
concentrations (see correlation coefficients in Table 3). The DI and the SDI, which are to some
extent interchangeable, allow a fine tuning of degradation intensities. The negative correlation
of the ox/anox ratio with the SDI is preconditioned, as it is the quotient of AA enriched by
degradation to those enriched in fresh plankton. It should be noted that this negative correlation
is even better than the positive correlation of the DI and the SDI. A close look at the SDI and
ox/anox shows that sediment samples from the regions with bottom water anoxia (Namibian
shelf < 200 m depths; Arabian Sea slope at 775 m) have lower ox/anox ratios and distinctly
higher SDI values compared with samples from similar depths and oxygenated bottom water
(e.g. Mediterranean Sea, Kara Sea, Eastern Pacific). The SDI and the ox/anox show a linear
correlation coefficient of R=-0.92 but the relationship is actually logarithmic (Fig. 7a). The SDI
better depicts the spectral changes in samples deposited under anoxic diagenetic conditions
such as those from the Namibian shelf (Nagel et al., 2016) and the Arabian Sea mid-water
oxygen minimum zone (Suthhof et al., 2001) while the ox/anox ratio better resolves variations
in samples of strong oxic degradation so that the SDI is in fact better suited to determine the
threshold of anoxic vs. oxic diagenesis. In anoxic sediments the SDI is close to the SDI in
sediment traps so that we surmise that anoxic diagenetic conditions preserve the primary SDI
signal from the water column (Fig. 7b). The SDI can thus be used to distinguish anoxic from
oxic diagenetic conditions in sediment cores. The SDI performs better than the ox/anox
indicator as it better resolves low values corresponding to low oxygen concentrations, and also
better than the DI as the latter less significantly correlates with the ox/anox indicator (Figure
S2). The good significant correlation of the SDI and POC (Table 3; R=0.57), furthermore,
implies that anoxic conditions favor the accumulation of POC or vice versa. It has been debated
whether sub- to anoxic diagenetic conditions enhance preservation as experiments showed
similar degradation rates at oxic and anoxic diagenetic conditions (Henrichs and Reeburgh,
1987). There is, however, ample evidence that POC contents are enhanced in anoxic sediments
so that the reduced proto- and metazoan grazing of bacteria by other organisms may be
responsible for the observed enhanced preservation (Lee, 1992).



Table 3: Pearson correlation coefficients of the SDI, RI, DI, Asp/β-Ala and Glu/γ-Aba with
selected AA*, the RTI, AAC/C, AAN/N, AA ratios and degradation indices, water depth
(Depth), POC and TN contents (%) and AA content (nmol/g) in sediment trap and sediment
samples (column 2-6). Pearson correlation coefficients of the RTI, RI, DI, Asp/β-Ala and Glu/γ-
Aba with selected AA, the SDI, AAC/C, AAN/N, AA ratios and degradation indices, water
depth (Depth), POC and TN contents (%) and AA content (nmol/g) in SPM samples (column
549 8-12).

* Only AA with a correlation coefficient R≥0.50 with at least one of the indicators are shown.

| 1 | 2 | 3 | 4 | 5 | 6 | 7 | 8 | 9 | 10 | 11 | 12 |
|---|---|---|---|---|---|---|---|---|---|---|---|
| Traps and Sediments | SDI | RI | DI | Asp/β-Ala | Glu/γ-Aba | SPM | RTI | RI | DI | Asp/β-Ala | Glu/γ-Aba |
| Asp | -0.68 | -0.30 | -0.83 | -0.25 | -0.32 | Asp | 0.81 | 0.00 | -0.61 | 0.13 | -0.17 |
| Thr | 0.12 | -0.25 | -0.25 | 0.03 | 0.07 | Thr | 0.86 | -0.02 | -0.57 | 0.06 | -0.16 |
| Ser | 0.32 | 0.03 | 0.16 | 0.09 | 0.16 | Ser | -0.91 | -0.15 | 0.08 | -0.11 | -0.01 |
| Gly | -0.21 | -0.32 | -0.45 | -0.31 | -0.31 | Gly | -0.90 | -0.17 | 0.12 | -0.11 | 0.04 |
| Ala | 0.46 | 0.07 | 0.17 | 0.03 | 0.11 | Ala | 0.91 | 0.09 | -0.43 | 0.11 | -0.09 |
| Val | 0.67 | 0.31 | 0.48 | 0.29 | 0.31 | Val | 0.88 | 0.07 | -0.26 | 0.08 | -0.09 |
| Ile | 0.87 | 0.48 | 0.76 | 0.52 | 0.55 | Ile | 0.68 | 0.34 | 0.26 | 0.26 | 0.02 |
| Leu | 0.88 | 0.49 | 0.76 | 0.52 | 0.56 | Leu | 0.65 | 0.47 | 0.28 | 0.35 | 0.11 |
| Tyr | 0.72 | 0.39 | 0.76 | 0.34 | 0.32 | Tyr | -0.14 | 0.27 | 0.66 | 0.02 | 0.23 |
| Phe | 0.89 | 0.38 | 0.79 | 0.40 | 0.44 | Phe | 0.78 | 0.37 | 0.04 | 0.29 | 0.06 |
| β-Ala | -0.85 | -0.27 | -0.44 | -0.32 | -0.35 | β-Ala | -0.08 | -0.66 | -0.19 | -0.52 | -0.12 |
| γ-Aba | -0.79 | -0.25 | -0.37 | -0.28 | -0.35 | γ-Aba | 0.24 | -0.53 | -0.65 | -0.20 | -0.40 |
| Lys | 0.56 | 0.14 | 0.43 | 0.15 | 0.15 | Lys | 0.71 | 0.05 | -0.20 | 0.03 | -0.06 |
| SDI | | 0.45 | 0.82 | 0.46 | 0.51 | SDI | 0.07 | 0.49 | 0.83 | 0.23 | 0.28 |
| RTI | 0.04 | -0.01 | -0.26 | 0.03 | -0.05 | RTI | | 0.10 | -0.34 | 0.12 | -0.08 |
| AAC/C% | 0.56 | 0.51 | 0.53 | 0.56 | 0.56 | AAC/C% | -0.08 | 0.14 | 0.22 | 0.06 | 0.09 |
| AAN/N% | 0.55 | 0.40 | 0.49 | 0.47 | 0.47 | AAN/N% | -0.05 | 0.06 | 0.03 | -0.06 | 0.08 |
| AA/HA | 0.54 | 0.58 | 0.54 | 0.73 | 0.75 | AA/HA | -0.57 | 0.11 | 0.32 | -0.01 | 0.16 |
| Gluam/Galam | 0.36 | 0.43 | 0.44 | 0.40 | 0.35 | Gluam/Galam | 0.27 | 0.43 | 0.16 | 0.28 | 0.08 |
| Asp/β-Ala | 0.46 | 0.93 | 0.57 | | 0.85 | Asp/β-Ala | 0.12 | 0.74 | 0.10 | | 0.07 |
| Glu/γ-Aba | 0.51 | 0.93 | 0.56 | 0.85 | | Glu/γ-Aba | -0.08 | 0.37 | 0.24 | 0.07 | |
| RI | 0.45 | | 0.49 | 0.93 | 0.93 | RI | 0.10 | | 0.33 | 0.74 | 0.37 |
| DI | 0.82 | 0.49 | | 0.51 | 0.56 | DI | -0.34 | 0.33 | | 0.10 | 0.24 |
| ox/anox | -0.91 | -0.30 | -0.71 | -0.31 | -0.37 | ox/anox | 0.67 | -0.19 | -0.50 | -0.10 | -0.09 |
| Depth | -0.67 | -0.28 | -0.54 | -0.28 | -0.37 | Depth | -0.55 | -0.12 | 0.23 | -0.11 | 0.04 |
| POC | 0.57 | 0.79 | 0.67 | 0.77 | 0.76 | POC | 0.27 | 0.30 | 0.13 | 0.25 | -0.03 |
| N | 0.45 | 0.46 | 0.50 | 0.39 | 0.39 | N | 0.29 | 0.30 | 0.13 | 0.24 | -0.01 |
| AA | 0.51 | 0.84 | 0.60 | 0.79 | 0.76 | AA | 0.17 | 0.28 | 0.15 | 0.20 | 0.01 |



The core SO90-111 KL from the oxygen minimum on the Pakistan margin was used to
reconstruct changes in oxygenation during the last 60 ka BP based e.g. on $\delta^{15}N$ values of total
N (Suthhof et al., 2001). The $\delta^{15}N$ values fluctuated between enhanced values in warm phases
due to denitrification in the mid-water oxygen minimum and lower values in cold phases when
the oxygen minimum zone was weaker or absent (Suthhof et al., 2001). The SDI very precisely
tracks these changes (Fig. 7c) and in accordance with the threshold discernable in Fig. 7b we
propose that the divide between oxic and anoxic diagenetic conditions is at SDI values between
0 and -0,2 with SDI<-0.2 indicating oxic and SDI >0 indicating oxic diagenetic conditions (Fig.
7a, b). The work of (Carr et al., 2016) - relying on the DI - suggests that signals of changes in
redox conditions can be preserved even down to 200 m core depth.



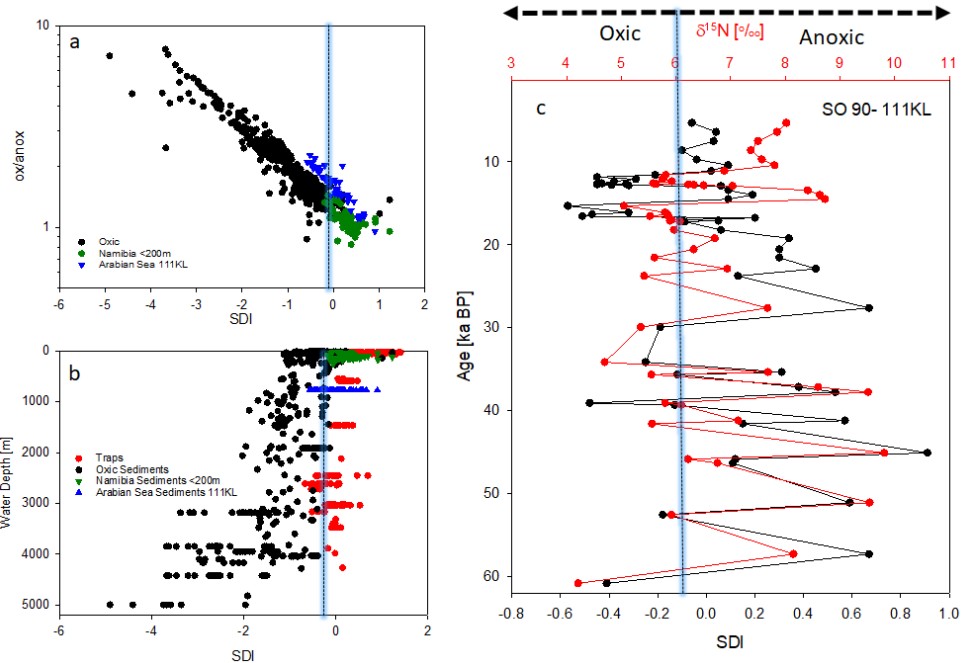


Figure 7: SDI indicator plotted against the log(ox/anox ratio) for oxic sediments (black) and
suboxic to anoxic sediments from Namibia (green) and the Arabian Sea (blue) (a). SDI plotted
with water depth (in m) of sediment trap deployment (red) and of sediment sampling (see color
code of a) (b). SDI and the $\delta^{15}$N of total N with sediment depth in sediment core SO90-111 KL
correlated with an $R^2$=0.48 (c); the blue bar marks the threshold of the SDI delimiting oxic and
anoxic diagenetic conditions at an SDI value of about -0.1.

571

**4.2.2 The RTI as an indicator of suspended matter residence time**

Changes in SPM are depicted by changes in RTI factor loadings (Fig. 5b) which are basically
due to the relative depletion of Phe, Ile, Leu, Glu and Tyr dominant in fresh plankton and the
enrichments of Gly, Ser, Orn and Lys. Both, Ser and Gly (Mol %) are strongly linearly
anticorrelated with the RTI (R=-0.91 and -0.90) showing that they can be used instead of the
RTI to determine the SPM residence time in the water column. The anticorrelation of the RTI
with water depth (Table 3) is due to the RTI decrease in the upper 200 m only, while it remains
constant below this depth (Fig. 8).

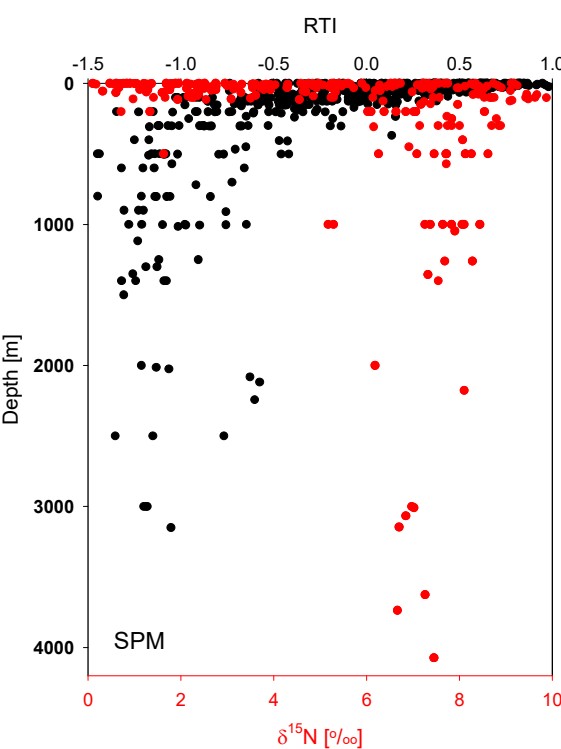

Figure 8: The RTI (black dots) and the $\delta^{15}N$ values (red dots) of SPM with water depths [m].

Below 200 m SPM becomes distinctly decoupled from sinking aggregates. SPM sampled in the
upper ocean mixed layer and euphotic zone includes fresh plankton whereas below the surface
mixed layer this fresh material is limited to the rare large sinking aggregates and the chance
that SPM interacts with sinking particles decreases with water depth due to the scarcity of both
(Mccave, 1984). The observed constant AA composition below 200 m water depth suggests
that SPM is largely refractory and the AA are barely accessible to further microbial degradation
or uptake. The age of the water masses in the upper ocean mixed layer is less than 100 years
while deeper waters have ages of several 100 years to maxima of 1600 years in the deep Indian
and Pacific Oceans (England, 1995; Gebbie and Huybers, 2012). Solubilization of particulate
matter by exoenzymes and the subsequent uptake in dissolved form (Carlson and Hansell, 2015;
Aristegui et al., 2009) leads to an almost complete turnover of originally diverse surface derived
organic matter. It is thus feasible that bacterial biomass comprises a large amount of organic
matter in compartments of long residence times. However, fresh bacteria and fungi have quite



similar AA composition as plankton (Cowie and Hedges, 1992) while SPM AA composition is
fundamentally different. The high AA/HA ratios associated with relatively high Gluam/Galam
ratios both not having a clear trend with water depth also suggest that the contribution of
bacterial biomass to SPM is small and does not increase with water depth (Table 1; Fig. 5a, b).
The observed changes in SPM are thus more likely related to adsorption processes and
macromolecule formation of material not digestible to deep sea organisms and resistant to their
enzymes. DOM was shown to become adsorbed to mineral surfaces (Keil and Kirchman, 1993;
Keil and Kirchman, 1994; Keil et al., 1994; Arnarson and Keil, 2005, 2007). However,
degradation of adsorbed AA proceeds on particles (Satterberg et al., 2003; Taylor, 1995). Thus,
the constant AA composition in SPM at depths >200 m may indicate that SPM is in equilibrium
with the TDAA which likewise show no depth dependent changes in AA composition (Figure
S3). Feasible candidate processes to explain the homogeneity are AA scavenging by SPM or
formation of gels (3D networks = biopolymers) which can anneal to larger sizes so that part of
the dissolved AA can be passed from the dissolved to the particulate organic carbon pool
(Druffel and Williams, 1990; Orellana and Leck, 2015). This process is, however, reversible so
that there is probably an exchange between the gel and particulate matter phase as well as
between gels of different sizes and complexities depending on pH, temperature, the presence
of ligands, pollutants or UV radiation (Orellana and Leck, 2015). Generally, hydrophobic AA
(Ala, Val, Met, Ile, Leu, Phe, Pro, Trp) and aromatic AA (His, Tyr) are more likely to form gels
and aggregates (Orellana and Leck, 2015). Our results indicate that an equilibrium may be
attained between the dissolved phase and SPM after a relatively short time so that the AA
composition of SPM is constant below 200 m water depth. If there is no further significant
scavenging of SPM by sinking particles and no degradation of AA on SPM, their abundance
could increase due to further adsorption of DOM. However, large zooplankters may be able to
utilize the SPM pool (Koppelmann et al., 2009; Gloeckler et al., 2018; Hannides et al., 2013)
and further studies are required to elucidate the fate of SPM in the ocean.
**5.3 Composition of total dissolved amino acids in sea water and pore water**
The uniform TDAA distribution with depth shows that most of the respiration is taking place
in shallow waters as is the case also for DOC (Reinthaler et al., 2006). After this instantaneous
utilization of AA by bacteria there are only small changes in the TDAA spectra (Figure S3) and
Ser, Gly and His uniformly become the major TDAA in sea water and pore water. It is possible
that the selective accumulation of these AA in the dissolved phase is due to their excretion or
their association with exoenzymes. Ser is present in N-acyl homoserine lactone (AHLs) which





is a class of bacterially produced signaling molecules involved in bacterial quorum sensing;
these compounds serve to regulate growth by changing gene expressions, for example, in order
to influence population density or phenotype (Parsek et al., 1999; Klein et al., 2009). His
changes from its protonated to deprotonated form at a pH of 6 and is therefore often present at
the active sites of enzymes. Ser and Gly may simply remain dissolved in sea water as they are
hydrophilic. Once mixed into the deeper ocean the scarcity of bacteria or the incorporation of
AA into gels could be the reason for their recalcitrance. However, we do not assume that a
considerable part of the TDAA belong to dissolved free AA. Because the differences between
the samples from different regions are much smaller than the difference between the molecular
weight fractions and sea water vs. pore-water (Figure S3), we surmise that the formation and
transformation processes of DOC are very uniform in the ocean. This assumption is based on
limited data so that these results are rather preliminary. We also do not have enough spatial
coverage of SPM and TDAA data in the deep ocean to detect AA utilization by organisms or
sorption and desorption processes. Both these organic matter pools are large (see below), so
that such investigations are important to estimate the possible role of these pools in oceanic
carbon sequestration and the reactions to global change (Ridgewell and Arndt, 2015).

**5.4 $\delta^{15}$N values in sinking and suspended matter and evidence for nitrogen sources and**
**transformation processes**
The $\delta^{15}$N values in sediments can preserve information on N sources throughout the geological
history (Sun et al., 2019; Gaye et al., 2018; Kienast et al., 2008). However, $\delta^{15}$N values may be
modulated by organic matter cycling and diagenetic processes which are replicated and thus
traceable in the AA composition not least because AA are the main identifiable contributors to
N. The increase of $\delta^{15}$N values by about 2 ‰ on average during organic matter burial and early
diagenesis in the upper sediments (Robinson et al., 2012; Tesdal et al., 2013) is corroborated
by a parallel shift in AA based degradation indicators (Gaye-Haake et al., 2005; Möbius et al.,
2010). In contrast to sediments, there are no clear depth related trends in $\delta^{15}$N values of sinking
particles in the water column of the epi- to mesopelagic ocean (Gaye-Haake et al., 2005; Yang
et al., 2017; Altabet, 2006). AA based biogeochemical indicators revealed degradation with
depth at specific trap sites (Haake et al., 1993a) and $\delta^{15}$N analyses of individual amino acids
showed that degradation is proceeding on sinking particles (Mccarthy et al., 2007) with $\delta^{15}$N
changes of "trophic" AA while $\delta^{15}$N of "source"AA remained constant (McCarthy et al., 2006).





However, degradation of sinking particles is much smaller than degradation at the sediment
water interface and in our large data set that integrates many different areas of study the small
to moderate changes in AA degradation are obviously obliterated, as neither AA contents (Fig.
3), nor the SDI (Fig. 7), the AAC/C % (Figure S4) nor AAN/N % (not shown) reveal any
significant trends in sinking particles in the deep ocean.
AA composition of SPM as expressed in the RTI is constant and SPM is rather recalcitrant at
water depths >200 m. Paralleling this, the $\delta^{15}$N values of SPM are about 6-8 ‰ on average in
all our studies carried out (Fig. 8). In previous studies $\delta^{15}$N values of SPM where reported to
increase from ≤5 ‰ in surface waters to values between 6-8 ‰ below 200 m water depth which
was attributed to organic matter degradation on SPM (Yang et al., 2017; Altabet et al., 1991;
Hannides et al., 2013; Emeis et al., 2010). However, SPM samples from the Arabian Sea
upwelling area show decreasing $\delta^{15}$N values from an average of 8.6 ‰ at water depth above to
7.4 ‰ at depths below 200 m (Gaye et al., 2013). It is thus reasonable that SPM has a constant
$\delta^{15}$N value in the mesopelagic and bathypelagic ocean. This is an additional indicator of a
common process determining the AA composition and their $\delta^{15}$N values of SPM and probably
also of DOM sampled below water depths of 200 m (equivalent to an age of ≥100 years;
(England, 1995; Gebbie and Huybers, 2012).

**5.5 Abundance of amino acids in the ocean**
Based on POC, TN and AA fluxes and the area of the open ocean and shallow seas (Costello et
al., 2010) we can estimate annual downward fluxes (Table S5). Average POC flux of
compilations of trap fluxes were between 1.65 g m$^{-2}$ a$^{-1}$ (Wilson et al., 2012) and 2.74 g m$^{-2}$ a$^{-1}$
(Rixen et al., 2019a) while our subset of trap samples from the open ocean (>2000 m water
depth) averages to 3.06 g m$^{-2}$ a$^{-1}$. For open ocean traps this results in total fluxes of 0.51-0.94
PgC a$^{-1}$. Our average flux estimates for TN are 0.13 PgN a$^{-1}$ and for AAC are 0.15 PgAA a$^{-1}$.
The flux rates over the shelves and slopes bear, however, large uncertainty because productivity
is by several orders of magnitude higher than in offshore areas and spatially variable. Our first
estimate, simply based on an average of our fluxes caught in traps deployed in areas of water
depth < 2000 m arrives at POC fluxes of 5.4 PgC a$^{-1}$, TN fluxes of 0.9 PgN a$^{-1}$ and AAC fluxes
of 1.36 Pg AAC a$^{-1}$. Thus 85-90 % of fluxes occur in near shore environments corroborating
that 95 % or the total marine organic carbon is buried in these environments (Hedges and Keil,





1995). The total sinking fluxes in the proximal plus distal ocean add up to 6.3 PgC a$^{-1}$, 1.0 PgN
a$^{-1}$ and, respectively, 1.51 Pg AAC a$^{-1}$ (see Table S5 for further details).
The largest organic carbon pool in the ocean is DOC with an inventory of 632±32 PgC (Carlson
and Hansell, 2015; Hansell et al., 2009) and the largest N pool is DON with 77±23 PgN (Gruber,
2008; Bronk, 2002). Dissolved AA are thus the largest AA pool in the ocean even if AA
comprise only a minor amount of DOC. We have only few measurements of AA concentrations,
which range between 0.1-0.2 mg/L with an average of 0.16 mg/L in all water samples excluding
bottom water. Based on these data we can estimate that AA comprise about 200±70 Pg which
would contribute about 35±11 Pg AAN and about 89±29 Pg AAC to the oceanic DON and,
respectively, DOC pools. Accordingly, AAC contributes about 14 % to DOC while AAN
contributes 45 % to total oceanic DON. This is in the low range of an estimate of 45-86 % AAN
based on NMR spectroscopy with acid hydrolysis recovering about half of this AAN pool
(Aluwihare et al., 2005).
The constant contents and composition of TDAA throughout the ocean indicates that it belongs
to the recalcitrant or refractory pool of DOC; this pool is hardly removed in the deep sea and
may only be degraded by photochemical reactions as it is returned into surface waters in the
course of ocean circulation (Legendre et al., 2015). Our TDAA data reveal no depth dependent
trend but our data coverage is not sufficient to detect any spatial variation. The distribution of
DOC is, however, well known with its maximum in surface water with 40-80 µmol C kg$^{-1}$ and
depletion in deep water with DOC concentrations from >50 µmol C kg$^{-1}$ in the North Atlantic
to 39 µmol C kg$^{-1}$ in the North Pacific deep water (Carlson and Hansell, 2015; Hansell et al.,
2009). Due to our limited number of measurements we may have missed spatial variations
which could elucidate TDAA sources and cycling processes in the ocean as is the case for DOC.
Respiration of DOC may be an important removal process in shallower waters (Reinthaler et
al., 2006) while a large proportion of the DOC reduction on its way to the Pacific on the deep
conveyer belt could be related to adsorption to POC, partly via gel formation (Druffel and
Williams, 1990).
TDAA may be among the constituents of DOC, which interact with SPM as both are transported
with their specific water masses by the ocean conveyer belt. Interaction with SPM is suggested
by the relative similarity in AA composition of TDAA and SPM. Moreover, SPM carries the
second largest pool of POC and AA in the ocean which has not been accounted for in carbon
budgets and which role in oceanic biogeochemical cycling has received little attention. The




total abundance of POC, TN and AA in SPM can be calculated using average concentrations
(Tab. 1) in the ocean volume between 0-200 m and between 200 m and the sea floor (Costello
et al., 2010). These calculations show that there are 443 Pg of total suspended matter in the
ocean of which organic carbon comprises 48 PgC, amino acids 35 PgAA and, total nitrogen 6
PgN. The relative similarity of AA spectra in SPM and TDAA suggests interaction between the
two pools at shallower depths and the build-up of an equilibrium, so that both pools remain
constant in concentrations and composition with depths. Like DOC, which was suggested to be
recalcitrant in the deep sea (Hansell and Carlson, 2013), SPM may only be affected by
degradation and repackaging into aggregates as it is reintroduced into surface water by ocean
circulation. Several studies, however, suggest that SPM may be an important food source for
deep living zooplankton (Koppelmann et al., 2009; Hannides et al., 2013; Gloeckler et al.,
2018). If there are no removal processes in the deep ocean, we would expect SPM and their
organic constituents to be exported from the Atlantic via the deep ocean circulation and to
accumulate in the Pacific.

**6 Conclusions**
The PCA of a set of 1425 samples consisting of sinking particle, SPM, sediment and water
samples produced two factors which separate AA in sinking particles and sediments on the one
hand from SPM and DOM on the other hand. As the PCA produced two branches diverging
with water and, respectively, sediment depth, strong interactions between the sinking and
suspended particles pools can be excluded.
The relative degradation of sinking particles and sediments, dominated by Gly, Asp, Glu and
Ala, can be tracked by a new degradation indicator named SDI derived from the first factor of
the PCA and correlated with the often-used degradation index DI. Except the SDI and the DI
all other biogeochemical indicators are not better than POC concentrations for a relative
classification of organic matter degradation. The SDI is, moreover, capable to separate oxic and
anoxic diagenetic conditions at an SDI between 0 to -0.2 (with values <-0.2 indicating oxic and
values >0 indicating anoxic diagenetic conditions). Application of the SDI furthermore, shows
that the diagenetic signal from the water column is preserved in sediments deposited under
anoxic conditions. The correlation of the SDI with POC shows that anoxic diagenesis enhances
POC accumulation in sediments compared to oxic diagenesis.



A novel biogeochemical indicator derived from the second factor of the PCA named RTI
depicts the transformation of SPM enriched in plankton derived AA in the epipelagic ocean to
a constant composition in the meso- and bathypelagic ocean. The deep SPM is probably the
residue of microbial processing and is not utilizable by enzymes under the present oceanic
conditions. This constant composition of SPM is corroborated by a constant $\delta^{15}N$ value below
200 m irrespective of the area of study.
DOM has constant AA composition throughout the water column, dominated by Ser, Gly, His,
Ala and Orn, pursuing the same accumulation AA pathway as found in deep SPM. Comparison
with literature data shows that the amount of AA released, depends on the intensity of
hydrolysis and that about 50 % of the amide linkages detectable by NMR spectroscopy cannot
be hydrolyzed. Similar to SPM the proteins are not utilizable by microorganisms. Protein-like
dissolved material was determined to be on average 2670 years old (Loh et al., 2004), showing
that these refractory molecules are cycled for several times before they can be removed by as
yet unknown processes.
Based on our AA data we have calculated the total oceanic AA inventory and found that TDAA
are the largest oceanic AA pool with a total amount of 200±70 PgAA and AA comprise 14 %
of the oceanic DOC and 45 % of oceanic DON.
The pool transported with SPM is 35 PgAA. SPM, furthermore, carries 48 PgC and 6 PgN not
accounted for in global carbon and nitrogen budgets. At present it is not known how the oceanic
DOM and SPM-particulate organic matter pool is formed and how this rather recalcitrant
organic matter can be removed from its abient water mass. It is feasible that these organic matter
pools have fluctuated in the past due to change in oceanic physicochemical conditions
(Ridgwell and Arndt, 2015). It is intriguing to understand how the accumulation or reduction
of this carbon and nitrogen pools has interacted with climate and environmental changes in the
geological history but it is vital to understand the response to ongoing and future climate
change.

**Data Availability**

Excerpts of the data were used in previous publications (i) from the Kara Sea in Gaye et al.
(2007) Nagel et al. (2005) and Unger et al. (2009), (ii) from the northern Indian Ocean in Gaye
et al. (2013), Gaye-Haake et al. (2005), Möbius et al. (2011) and Suthhof et al. (2001), (iii) from



the Mediterranean Sea in Möbius (2013) and Möbius et al. (2010), (iv) from the Namibian
upwelling in Nagel et al. (2016) and (v) from the Pacific in Paul et al. (2018). The entire data
set will be made available in PANGAEA. Data from the Pacific are available at:
https://doi.pangaea.de/10.1594/PANGAEA.885391,  https://doi.pangaea.de/10.1594/PANGAEA.881804,
https://doi.pangaea.de/10.1594/PANGAEA.881813.

**Acknowledgements**
This work is based on samples taken during cruises of research vessels R/V SONNE, METEOR,
MARIA S. MERIAN, PELAGIA, ORV SAGAR KANYA and R/V AKADEMIK PETROV.
We are grateful to all officers of crew of these research vessels. We are indebted to Venugopalan
Ittekkot who initiated and inspired this work and led many of the research projects. We thank
Desmond Gracias, Areef Sardar and Fernando Vijayan from NIO, Goa, India for technical
support on board. We thank Inken Preuss, Annika Moje, Tim Jesper Suhrhoff and Seinab
Bohsung for help with pore water sampling during the cruises SO239 and SO242. We are
indebted to Frauke Langenberg and Marc Metzke for their high-quality analyses. We thank the
German Federal Ministry of Education and Research for funding (grant no.: 03F0707G) in the
framework JPI Oceans EcoMining-DEU - Ecological Aspects of Deep-Sea Mining to Jacobs
University Bremen. Sampling in the southern Indian Ocean was conducted within the
framework of the INDEX program of the Federal Institute for Geosciences and Natural
Resources (BGR). The DFG and BMBF financed the finalized or terminated projects in the
northern Indian Ocean and Atlantic (Indian-German Program, JGOFS, BIGSET), the
Mediterranean Sea (MEDNIT), the Namibian upwelling (GENUS) and the Kara Sea (SIRRO).

**Author Contribution**
BG, NL, TR and KE designed the study and led the projects in which samples were taken and
analyzed. NL developed and refined the AA analyses. NL, NH and SP contributed and analyzed
samples from the southern Indian Ocean and the Pacific. BG wrote the manuscript with
contributions of all co-authors.




**Competing interests**
The authors declare that they have no conflict of interest.

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
