# Peer review of "What can we learn from amino acids about oceanic organic matter cycling"

_Biogeosciences, 2021_

## Author Comment (AC1)

In this manuscript, Gaye and colleagues measure amino acid (AA) concentration in a large sample set including particulate, sedimentary, and dissolved organic matter to assess the utility of AA molar abundance-based degradation proxies. Based on trends in AA molar abundance, they suggest suspended and particulate OM undergo separate degradation pathways, and that current degradation indices do not function as expected for suspended particulate matter (SPM). They suggest two new indices which can be calculated from AA molar abundance. The first, the sediment degradation indicator (SDI) is suggested as an alternative to the degradation index (DI) for sinking particles and sediments. The second, the residence time indicator (RTI) is proposed as an indicator specific to the degradation of SPM. Clearly, a lot of work went into the impressive dataset presented in this manuscript. I believe the authors' exploration of existing AA-based degradation proxies as well as the introduction of new proxies is of interest to the wider biogeochemistry community. However, I feel there are some issues that should be addressed prior to publication, including tightening of the introduction, clarification of some methodologies, and providing additional support for some conclusions. Below are my specific suggestions.

*Reply: We thank the referee for taking the time to review our manuscript and for the detailed comments which will help to improve the manuscript. The referee has indeed identified the points which need to be specified, further discussed or corrected.*

General comments:

- DI application to SPM and DOM: The authors note that the classic calculation of DI according to Duawe et al., 1999 does not seem applicable to SPM and DOM. However, there is an alternative DI calculation suggested by Kaiser and Benner 2009 specific for DOM. This calculation is more appropriate for DOM samples and based on their argument that SPM cycling is more similar/linked to DOM cycling than suspended particles, may also be more appropriate for their SPM samples. I suggest the authors add this calculation to their analyses before arguing existing DI calculations are not appropriate for their sample set. Comparisons with the RTI should also be included.

  *Reply: we already approached the authors and will calculate their DOM specific DI and check its suitability for our data set and also compare it to our RTI.*

- "Residence time" terminology: I'm not convinced the RTI is an indicator of "residence time", as suggested by the name and by section header 4.2.2. All the RTI indicates is changes to AA molar abundance. Throughout section 4.2.2, residence time is only mentioned once (lines 589-591). Instead, most of this discussion focuses on hypothesized relationships between SPM and DOM. Additionally, as the authors note, water mass age can vary significantly below 200 m (line 590), while the RTI is relatively constant at these depths. If the authors want to claim that the RTI is an indicator of residence time, I think they need to make a clearer connection/stronger argument in this section. Otherwise, a connection to hypothesized degradation seems more consistent with their data.

  *Reply: The connection of changes in the "RTI" with residence time is indeed speculative. It is based on the observation that samples from water depths below about 200 m show no depth dependent trend in AA composition while there is a trend of increasing "RTI" in the upper 200 m. The referee is right that according to our theory the equilibrium with ambient DOM drives the changes and is evidently taking place relatively fast (about 100 years or faster according to water*

*mass ages and the fact that fresh organic matter is constantly supplied/produced in near surface water). We will think about a better term for this SPM-DOM specific new index and also work on its discussion. Checking the DI of Kaiser and Benner 2009 may also help here.*

- Range of sampling locations: One strength of this paper is the very large dataset they use for their analyses. However, their samples come from a very wide range of sampling environments. The authors do mention that there is more variation between sample types than between similar sample types from different regions (lines 460, 637), but this appears to be almost an afterthought. I think it would be helpful if a brief description of variation within each sample type between the different locations was presented in the results and/or earlier in the discussion.

*Reply: we will include a section on the different sample times and briefly about the possible variations between working areas.*

Specific comments:

Figures:

Overall: The authors are inconsistent with their use of identifying colors/symbols/etc in figure captions. Per the journal guidelines, "A legend should clarify all symbols used and should appear in the figure itself, rather than verbal explanations in the captions (e.g. "dashed line" or "open green circles")."

*Reply: We will check all Figures and supply legends instead of descriptions in the text.*

Figure 3: A legend should be included for the red vs. black symbols.

*Reply: legend will be included in the Figures.*

Figure 4: This figure seems to have some repeated information and unclear legend entries. The caption says 4b compares AA mol% of "plankton, SPM, and water samples", while figure legend says "dissolved" (instead of water samples). For 4c, the caption says, "water samples and pore water", but here the legend says "water" instead of "dissolved" or "water samples." The authors need to be clearer about dissolved vs. water vs. pore water (does "dissolved" include water column and pore water combined?). Also, the caption notes what the colors mean for part c, but not for parts a and b. Finally, plankton data is presented in both panels a and b, and it is unclear if there is repeated water column data in b and c. While repeating some data in multiple panels allows direct comparison between certain sample types, it appears the discussion text only directly compares sediment traps and SPM (line 439), which are noton the same plot in any panels. Would it be possible to remove repeated info and condense the figure to two panels? Finally, including an asterisk for significant differences might help aid the eye to see which differences are important.

*Reply: We will add the explanations of colours of Fig.4a and b into the caption. Fig. a and b both, contain plankton sample composition. Plankton is shown in duplicate in order to clarify the trends from plankton via trap samples to sediments as well as from plankton to SPM and further to DOM. However, several of the remarks of the referee ask us to work on the trends observed in SPM. We will check if splitting SPM into shallow and deep samples would make more sense and in case show this in Figure 4b. Figure 4c splits the dissolved AA results into samples from the water column and from pore waters. Both groups are indeed quite similar in composition so that 4c could be deleted, especially as further and more detailed information on the differences between these two groups of samples is provided in supplement S3.*

*We will also add asterisks to indicate AA which show changes to aid the eye looking at many columns.*

Figure 5c: It appears two or three of the box and whisker plots are cut off.

*Reply: We will redraw Figure 5a and c.*

Figure 6a: would it be possible to separate the overlapping amino acid labels to improve readability? (Perhaps with an arrow pointed to exactly where the factor loadings are for that AA).

*Reply: We will separate the overlapping labels according to the suggestion.*

Figure 7: The text in most of the figures is small, but the text in 7a and b is so small it is almost illegible. It is also inconsistent with the text size in 7c. Additionally, I suggest the authors include the regression line in figure 7a.

*Reply: We will increase the font of the text in this Figure and also check the other Figures with small texts. Regression line will be included in Fig. 7a.*

Figure S2: Could the regression lines be plotted on these figures? It's possible part of the regression line can be seen in figure S2a, though if this is the case it is mostly hidden by the data points in the same column. Perhaps a separate color could be used?

*Reply: We will add regression lines in different colours.*

Text:

Line 112 (and other places throughout the manuscript): change citation to say "McCarthy"

*Reply: Will be changed.*

Lines 117-124: Considering the authors do not use/test different hydrolysis conditions, this seems like a lot of unnecessary detail in an already long introduction.

*Reply: Will be deleted here and moved partly to the discussion.*

Lines 126-128: This is methods text, it does not belong in the introduction.

*Reply: Will be moved to methods*

Section 2. This section feels like a long mix of introduction, discussion, and methods. I think in general the whole introduction could be shortened. One way to do this which may improve readability is to move the calculations for each index to a separate methods subsection and only include a concise summary of different in the introduction. Any remaining details necessary for the discussion of results could be moved to the relevant discussion sections (some of which are repeated in the discussion anyway).

*Reply: Calculation methods of indicators will be moved as a much smaller section to the methods. Other parts will be moved to the discussion where appropriate.*

Line 170: Conversion of Asp and Glu via hydrolysis is methods text (and is repeated in the methods). Should be deleted here.

*Reply: Will be deleted*

Line 190: This additional discussion of Asp and Glu seems to come out of nowhere. I would consider including this with the previous paragraph about Asp Glu.

*Reply: Will be deleted*

Line 217: As noted above, a separate DOM-specific calculation was suggested which is more applicable to DOM.

*Reply: See reply to the first general comments above; we will calculate the DOM specific degradation index of Kaiser and Benner (2009) and then compare it with our "RTI" and decide which index is best applicable.*

Lines 252, 284: While the authors mention use of Whatman GF/F filters, it might be helpful for readers to also provide the pore size rather than assuming they will know. Overall, I think a clear size range for each sample type would be helpful (similar to that provided in lines 254-255 for sediment trap samples).

*Reply: We will add the pore size of these glass microfiber filters of 0.7 $\mu$m.*

Line 258-261: Based on this text, it appears the only water sampling was pore-water samples and 18 water samples off Namibia. However, Figure S3 implies there were additional water column samples collected not mentioned here. If so, the authors should describe those sampling procedures here as well. Additionally, it would be helpful if the authors provide a pore size for the rhizon samplers.

*Reply: We will add information on the water samples which are from the southern Indian Ocean and from the central Pacific.*

Line 370-370: "particles and sediments have increasing mol%..." Increasing with what? Greater than plankton samples? Or is this meant to reflect some relationship with depth or some other variable?

*Reply: This is indeed imprecise. We will clarify that this trend of increasing mol% is depth dependent.*

Line 387: This sentence is hard to follow. It begins with a comparison between SPM and sed trap samples, but rather than giving any data for sed trap samples transitions to comparing AA/HA in shallow and deep SPM with Gluam/Galam ratios in SPM.

*Reply: We will rewrite the sentence and explain it better.*

Line 397: "significantly different" implies a statistical difference. If this is what the authors mean, the statistics should be included.

*Reply: The significance could be tested by a discriminant analysis which is too much effort. We will instead remove "significantly" and refer to the close mean values and overlapping standard deviations (Fig. 5c and d).*

Line 436: "shares" is a confusing term here. Do the authors mean mol%?

*Reply: Will be changed.*

Line 436-437: There is no depth data in figure 4b to support this claim.

*Reply: We will change this statement. In Figure 4b only the trends from plankton via sinking particles to sediments are visible while the trends within the sample groups are not shown in this paper. We will rethink how to refer to the trends within the sample groups in this paper. This is related to some of the earlier remarks (see above).*

Line 443: This feels like a jump in logic to me. The authors note at the end of the results that most calculated indices don't show major differences in SPM with depth, but I think they need to expand on this in discussion prior to making the claim that SPM has a different degradation pathway that is not captured by these indices. Especially because many of the AAs which are enriched with water depth in SPM (Gly, beta-Ala, gamma-Aba, Orn) are also enriched with depth in sediments.

*Reply: beta-Ala and gamma-Aba are not enriched in SPM with depth which is one of the reasons why we think that it is not the classical degradation which drives changes in SPM. Anyway, this discussion will be expanded in the revised version (see comments and replies above).*

Line 490: There is no information regarding depth in Fig. 6. This can only be seen in figure S1.

*Reply: We will change this and refer only to S1.*

Line 505: The topic sentence for this paragraph suggests there will be discussion regarding if individual AA can be used in place of SDI and RTI, but there is no further mention of this in section 5.2.1 (though there is section 5.2.2).

*Reply: In line 576 (section 5.2.2) we mention that mol-% Gly or Ser could be used instead of the RTI to characterise the relative changes in suspended matter. We find this important as several groups do not measure beta-Ala and gamma-Aba which are needed to calculate our SDI and RTI.*

Line 524: is this also referencing figure S2?

*Reply: A reference to S2 will be added.*

Lines 525-530: There appear to be three separate arguments in this sentence, but the overall reasoning/data to support these arguments is unclear to me. Could the authors break this down, so each argument is presented separately with the data to support it? I think would improve readability and help convince readers that SDI is in fact an improvement over the DI and ox/anox ratio.

*Reply: Thank you for this comment. The three arguments will be presented separately in a more logical order.*

Line 651-652: AA are not the main identifiable contributors to all global nitrogen- a qualifier, such as sedimentary or particularly organic N is necessary.

*Reply: We will add that this is the case in most particulate organic matter samples.*

Line 660: I believe this is referencing the same paper as the line above (McCarthy 2007), as there is no McCarthy 2006 paper is in references. The citation should be corrected and probably only needs to be cited once at the end of the sentence.

*Reply: This reference will be corrected.*

Line 672: should this say water depth above 200m?

*Reply: Yes, the 200 m will be added.*

Line 698: Can the authors discuss how these calculations compare with past work and direct measurements of %OC and ON which is AAs? For instance, Bronk 2002 suggests dissolved combined AAs only represent ~ 7% of total DON, and Kaiser and Benner 2009 suggest amino acids, amino sugars, and neutral sugars collectively account for 1% to 5% of TOC in the Pacific and Atlantic gyres. Perhaps their values are higher because most of their sampling locations are coastal rather than the open ocean?

*Reply: Lines 113-124 contain some more information on % AAN of N which will be moved to the Discussion. The above mentioned references will also be added.*

Line 748: I think it would be helpful to mention in paratheses which biogeochemical indicators are not better than POC, as there are others which are not investigated in this study.

*Reply: The indicators not better than POC % (and DI and SDI) used in this work will be added.*

---

## Author Comment (AC2)

Though amino acid composition has been regularly measured as an indicator of organic matter degradation, the varied sources and processes beyond heterotrophic degradation shift the AA composition and limited its application in diverse marine environments. It has been frequently found in recent publication that the DI index was inappropriately applied to samples that were clearly regulated by mechanisms different from Dauwi's dataset. However, the manuscript by Gaye et al. analyzed amino acid composition from amazingly abundant samples covering traps, sediment, suspended particles, the seawater and pore water from varied regions. This provides a complete dataset to reliably reflect the variation in amino acid composition among major pools of amino acids in the ocean. This work well fits the scopes of Biogeoscience. The manuscript was well written and the conclusion is convincing and informative. I would suggest the publication of this work with some minor editions as followings.

*Reply: We thank the reviewer for taking the time to review our paper and for the very encouraging words.*

*We plan to respond to the specific comments in the following way:*

**Specific comments:**

1. Considering the spatial coverage and large range of concentrations, whether it is suitable to calculate the decay functions (Fig. 3) using the entire dataset?

*Reply: In Figure 3 we excluded Kara Sea samples as resuspended sediments was found to impact SPM and trap samples as we published Gaye et al. 2007. In a revised version we will do both calculations and check if it makes sense to use the entire data set or again exclude Kara Sea samples due to the bias.*

2. Were Gluam and Galam measured simultaneously with the amino acids?

*Reply: They were measured simultaneously with amino acids and we will clarify this in the methods section of the revised version.*

3. One thing confused me a lot is where comes the amino acids enriched in pore water, if there is no exchange between pore water and sediment.

*Reply: We will check the respective part of the manuscript and explain it in a better way and in more detail.*

*The following changes will be made*

4. Line 268, "carbon and N" to "carbon and nitrogen"

5. Line 406 "Least degraded" to "most degraded"?

6. Line 436, "increasewith" to "increase with"

7. Line 725, "Tab. 1" to "Table. 1"

---

## Author Response (AR1)

**Reply to Reviewer 1:**

In this manuscript, Gaye and colleagues measure amino acid (AA) concentration in a large sample set including particulate, sedimentary, and dissolved organic matter to assess the utility of AA molar abundance-based degradation proxies. Based on trends in AA molar abundance, they suggest suspended and particulate OM undergo separate degradation pathways, and that current degradation indices do not function as expected for suspended particulate matter (SPM). They suggest two new indices which can be calculated from AA molar abundance. The first, the sediment degradation indicator (SDI) is suggested as an alternative to the degradation index (DI) for sinking particles and sediments. The second, the residence time indicator (RTI) is proposed as an indicator specific to the degradation of SPM. Clearly, a lot of work went into the impressive dataset presented in this manuscript. I believe the authors' exploration of existing AA-based degradation proxies as well as the introduction of new proxies is of interest to the wider biogeochemistry community. However, I feel there are some issues that should be addressed prior to publication, including tightening of the introduction, clarification of some methodologies, and providing additional support for some conclusions. Below are my specific suggestions.

*Reply: We thank the referee for taking the time to review our manuscript and for the detailed comments which will help to improve the manuscript. The referee has indeed identified the points which need to be specified, further discussed or corrected.*

General comments:

- DI application to SPM and DOM: The authors note that the classic calculation of DI according to Duawe et al., 1999 does not seem applicable to SPM and DOM. However, there is an alternative DI calculation suggested by Kaiser and Benner 2009 specific for DOM. This calculation is more appropriate for DOM samples and based on their argument that SPM cycling is more similar/linked to DOM cycling than suspended particles, may also be more appropriate for their SPM samples. I suggest the authors add this calculation to their analyses before arguing existing DI calculations are not appropriate for their sample set. Comparisons with the RTI should also be included.

*Reply: The Indicator developed by Kaiser and Benner (2009) displays a depth dependent trend at one of their sampling stations while this indicator is almost constant with depth at the other – the latter is similar to our finding. Unfortunately, we did not receive the data to calculate the DOM specific degradation index from the author. We searched for it in all the papers cited in Kaiser and Benner (2009) and they are not available so that we cannot use their Index. However, our DOM data show no trend with depth in the water column in none of the common indicators so that we don´t expect this from a new indicator derived from a set of DOM samples.*

*But the paper which we overlooked in the initial version of our manuscript (although several papers of the group were cited) has turned out to support the utility of our RTI as Kaiser and Benner find a similar relationship between ventilation ages of water masses as we suggest based on the general concept of water mass ages In the ocean. Kaiser and Benner have very detailed data of water mass ages at their long-term stations HOTS and BATS. Using different indicators for AA degradation they come to the same results as displayed by the RTI (lines 620-622).*

- "Residence time" terminology: I'm not convinced the RTI is an indicator of "residence time", as suggested by the name and by section header 4.2.2. All the RTI indicates is changes to AA molar abundance. Throughout section 4.2.2, residence time is only mentioned once (lines 589-591). Instead, most of this discussion focuses on hypothesized relationships between SPM and DOM. Additionally, as the authors note, water mass age can vary significantly below 200 m (line 590), while the RTI is relatively constant at these depths. If the authors want to claim that the RTI is an indicator of residence time, I think they need to make a clearer connection/stronger argument in this section. Otherwise, a connection to hypothesized degradation seems more consistent with their data.

*Reply: We found more evidence in the literature that the residence or renewal time of the water mass is indeed an important factor determining the organic matter degradation state displayed in AA composition (see comment above). We have specified in the Results and the Discussion that the patterns of relative AA loss and enrichment in SPM do not correspond with the patterns known to present AA degradation processes and depicted by common biogeochemical indicators. A summary paper by Lam and Marchal (2005) based on a large data set supports that there is decreasing exchange of sinking particles and SPM while the main interaction in the meso- and bathypelagic ocean is between SPM and DOM.*

*We now discuss possible reasons for the fact that the AA spectra of SPM do not change with depth below 200m and that the organic matter is either recalcitrant or that it cannot be utilized due to the low concentrations of SPM in analogy to the dilution theory of Arrieta et al. (2015).*

*In summary, we have improved results and discussion, added supporting literature that the RTI can be used to characterize SPM. The pattern the RTI shows is consistent with the literature and comes to similar results as studies on SPM using e.g. thorium isotopes, radiocarbon and biomarkers. We think that the term "RTI" which can stand for both, refractivity or residence time index is indeed useful based on the present state of knowledge.*

- Range of sampling locations: One strength of this paper is the very large dataset they use for their analyses. However, their samples come from a very wide range of sampling environments. The authors do mention that there is more variation between sample types than between similar sample types from different regions (lines 460, 637), but this appears to be almost an afterthought. I think it would be helpful if a brief description **of variation within each sample type between the different locations** was presented in the results and/or earlier in the discussion.

*Reply: In the methods we now briefly describe the samples taken in the different regions (lines 159-175). Preparing the review we double checked the results from the different study areas could thereby confirm that the differences are rather small and not systematic between samples of the same "type" and we have decided not to show these data. Again checking the entire data set made us change Figure 4. AA composition of trap samples can be averaged for the entire water column and the relatively small differences between shallow and deep trap samples are evident from the biogeochemical indicators shown in Figure 5 and Table 1. However, the AA composition of SPM is considerably different in shallow (<200 m) and deep waters and this is therefore presented in Figure 4b. The discussion starts with the statement that the difference between the samples from different areas and we have modified the sentence to make this clearer (lines 439-443):*

"Our summary of AA data from various locations in the world ocean corroborates earlier findings that degradation of organic matter by zooplankton and microbes imparts

characteristic changes to AA spectra so that the differences in AA composition are much larger between types of organic matter and from different water and sediment depths than between different oceanic areas (Lee, 1988)."

Specific comments:

Figures:

Overall: The authors are inconsistent with their use of identifying colors/symbols/etc in figure captions. Per the journal guidelines, "A legend should clarify all symbols used and should appear in the figure itself, rather than verbal explanations in the captions (e.g. "dashed line" or "open green circles")."

*Reply: All Figures were checked and legends were supplied in addition to descriptions in the text.*

Figure 3: A legend should be included for the red vs. black symbols.

*Reply: Legends are now included in all Figures explaining the symbols.*

Figure 4: This figure seems to have some repeated information and unclear legend entries. The caption says 4b compares AA mol% of "plankton, SPM, and water samples", while figure legend says "dissolved" (instead of water samples). For 4c, the caption says, "water samples and pore water", but here the legend says "water" instead of "dissolved" or "water samples." The authors need to be clearer about dissolved vs. water vs. pore water (does "dissolved" include water column and pore water combined?). Also, the caption notes what the colors mean for part c, but not for parts a and b. Finally, plankton data is presented in both panels a and b, and it is unclear if there is repeated water column data in b and c. While repeating some data in multiple panels allows direct comparison between certain sample types, it appears the discussion text only directly compares sediment traps and SPM (line 439), which are not on the same plot in any panels. Would it be possible to remove repeated info and condense the figure to two panels? Finally, including an asterisk for significant differences might help aid the eye to see which differences are important.

*Reply: Figure 4 was redrawn to avoid redundant data presentation. Further, a legend was added explaining all colours. Fig. 4a was kept while Figure 4b was changed to shallow and deep SPM and total dissolved amino acids in water samples. This allows to discuss changes in SPM AA composition with depths. Figure 4c was deleted and the resemblance of water and pore water spectra was only mentioned in the text and further and more detailed information on the differences between these two groups of samples is provided in supplement S3.*

*We added asterisks to indicate AA which increase within the sample groups displayed in the Figure to aid the eye looking at many columns.*

Figure 5c: It appears two or three of the box and whisker plots are cut off.

*Reply: Figure 5a and c were redrawn and all boxes and whiskers are shown, some of the outliers are cut off for better perceptibility of the trends and differences between the groups of samples and this is mentioned in the Figure captions.*

Figure 6a: would it be possible to separate the overlapping amino acid labels to improve readability? (Perhaps with an arrow pointed to exactly where the factor loadings are for that AA).

*Reply: Overlapping labels were changed and arrows inserted.*

Figure 7: The text in most of the figures is small, but the text in 7a and b is so small it is almost illegible. It is also inconsistent with the text size in 7c. Additionally, I suggest the authors include the regression line in figure 7a.

*Reply: We increased the font of the text in this Figure and also changed all other Figures with small fonts. Regression line of the exponential regression for all sediment samples is included in Fig. 7a; this regression is identical to the regression in Figure S2a. In order to be consistent both Figures now have a linear scale (Fig. 7a was changed) and the identical exponential regression with an R=-0.95.*

Figure S2: Could the regression lines be plotted on these figures? It's possible part of the regression line can be seen in figure S2a, though if this is the case it is mostly hidden by the data points in the same column. Perhaps a separate color could be used?

*Reply: We added regression lines in different colours and provide the correlation coefficients of the best correlation which is an exponential correlation.*

Text:

Line 112 (and other places throughout the manuscript): change citation to say "McCarthy"

*Reply: Changed.*

Lines 117-124: Considering the authors do not use/test different hydrolysis conditions, this seems like a lot of unnecessary detail in an already long introduction.

*Reply: This was deleted here and moved to the discussion of DOM results (lines 677-686).*

Lines 126-128: This is methods text, it does not belong in the introduction.

*Reply: Was moved to methods (lines 206-208)*

Section 2. This section feels like a long mix of introduction, discussion, and methods. I think in general the whole introduction could be shortened. One way to do this which may improve readability is to move the calculations for each index to a separate methods subsection and only include a concise summary of different in the introduction. Any remaining details necessary for the discussion of results could be moved to the relevant discussion sections (some of which are repeated in the discussion anyway).

*Reply: Calculation methods of indicators was moved to the methods as a separate chapter (lines 235-301). It is basically a shortened version with the same header as the previous section 2 but restricted to the calculation and application of amino acid based biogeochemical indicators. A very short paragraph describing the use of amino acid derived biogeochemical indicators was instead retained in the introduction (lines 208-123).*

*The entire paragraph on hydrolysis and AA yields from hydrolysis was moved to the discussion (see above; lines 677-686)*

*Other parts will be moved to the discussion where appropriate.*

Line 170: Conversion of Asp and Glu via hydrolysis is methods text (and is repeated in the methods). Should be deleted here.

*Reply: Deleted*

Line 190: This additional discussion of Asp and Glu seems to come out of nowhere. I would consider including this with the previous paragraph about Asp Glu.

*Reply: The section on Asp, Glu and their degradation products was moved to the methods (lines 226-228) and shortened in order to avoid redundance.*

Line 217: As noted above, a separate DOM-specific calculation was suggested which is more applicable to DOM.

*Reply: See reply to the first general comments above; we have included a comparison with the results of the paper and this supports our ideas but we did not receive the results of the PCA and thus could not test this indicator.*

Lines 252, 284: While the authors mention use of Whatman GF/F filters, it might be helpful for readers to also provide the pore size rather than assuming they will know. Overall, I think a clear size range for each sample type would be helpful (similar to that provided in lines 254-255 for sediment trap samples).

*Reply: We added the pore size of these glass microfiber filters of 0.7 μm and clarified that this is the distinction between dissolved samples and SPM similar to the description of the pore size for filtration of sediment trap samples (lines 176-177; 186).*

Line 258-261: Based on this text, it appears the only water sampling was pore-water samples and 18 water samples off Namibia. However, Figure S3 implies there were additional water column samples collected not mentioned here. If so, the authors should describe those sampling procedures here as well. Additionally, it would be helpful if the authors provide a pore size for the rhizon samplers.

*Reply: This section was modified in order to clarify how water samples were taken and that the 18 samples are only part of the water samples analysed (lines 176-181):*

"Sea water was filtered through glass fiber filters (Whatman GF/F) with a nominal pore size of 0.7 μm and filters were dried at 40°C in order to obtain SPM samples. At some stations water samples were taken by deep freezing an aliquot of the filtrate for dissolved AA analyses. In addition 18 water samples taken off Namibia were separated into two size classes by ultrafiltration."

*The mean pore size of rhizones of 0.15 μm was added (line 186).*

Line 370-370: "particles and sediments have increasing mol%..." Increasing with what? Greater than plankton samples? Or is this meant to reflect some relationship with depth or some other variable?

*Reply: This is indeed imprecise. We clarified that this trend of increasing Mol% is depth dependent (lines 371-392).*

Line 387: This sentence is hard to follow. It begins with a comparison between SPM and sed trap samples, but rather than giving any data for sed trap samples transitions to comparing AA/HA in shallow and deep SPM with Gluam/Galam ratios in SPM.

*Reply: The whole part on AA/HA and Gluam/Galam was rewritten and the observations are presented more clearly and simplified (lines 401-407).*

Line 397: "significantly different" implies a statistical difference. If this is what the authors mean, the statistics should be included.

*Reply: We removed "significantly" and referred to the close mean values and range of values (Fig. 5c and d) (line 412/413).*

Line 436: "shares" is a confusing term here. Do the authors mean mol%?

*Reply: Mol% was added.*

Line 436-437: There is no depth data in figure 4b to support this claim.

*Reply: Figure 4 was redrawn. It was restricted to 4a and b and SPM shallow and deep samples were separated in 4b so that this claim can be made from the Figure.*

Line 443: This feels like a jump in logic to me. The authors note at the end of the results that most calculated indices don't show major differences in SPM with depth, but I think they need to expand on this in discussion prior to making the claim that SPM has a different degradation pathway that is not captured by these indices. Especially because many of the AAs which are enriched with water depth in SPM (Gly, beta-Ala, gamma-Aba, Orn) are also enriched with depth in sediments.

*Reply: The differences between the enrichment patterns in SPM and sinking particles are described much clearer now: beta-Ala and gamma-Aba are not enriched in SPM with depth and Asp even becomes depleted. These are reasons why the Asp/ß-Ala, Glu/g-Aba, RI do not work with SPM. The DI in principle also summarizes the degradation pattern and does not work either. Some of the prominent changes in SPM with depth and the prominent difference between TDAA and the other types of samples such as the enrichments of Ser, His and Orn are not depicted in the DI or even with different prefix and there are no other biogeochemical indicators based on these AA.*

*We rewrote the text and made the difference much clearer now and hope that it is now more convincing (lines 387-403).*

Line 490: There is no information regarding depth in Fig. 6. This can only be seen in figure S1.

*Reply: Figure 6 would be very confusing with signatures referring to water depths. We used biogeochemical indicators in Figure 5 and refer to S1 where samples from <200m and >200m are shown by different colours.*

Line 505: The topic sentence for this paragraph suggests there will be discussion regarding if individual AA can be used in place of SDI and RTI, but there is no further mention of this in section 4.2.1 (though there is section 4.2.2).

*Reply: In line 576 (section 4.2.2) we mention that mol-% Gly or Ser could be used instead of the RTI to characterise the relative changes in suspended matter. We find this important as several groups do not measure beta-Ala and gamma-Aba which are needed to calculate our SDI and RTI. In order not to focus too much on this topic we deleted the hint in line 505 but retained the results, that Gly and Ser MOl% are suitable indicators in section 4.2.2 (lines 597-599).*

Line 524: is this also referencing figure S2?

*Reply: A reference to S2 was added. Also, Figure 7a was changed to a linear scale (see reply to Figure 7 above).*

Lines 525-530: There appear to be three separate arguments in this sentence, but the overall reasoning/data to support these arguments is unclear to me. Could the authors break this down, so each argument is presented separately with the data to support it? I think would improve readability and help convince readers that SDI is in fact an improvement over the DI and ox/anox ratio.

*Reply: Thank you for this comment. In order to explain Figure 7a-c better we rewrote this part of the discussion. It is much clearer now why the SDI is an improvement over the DI and the ox/anox. We decided to delete the discussion on the link between oxia vs. anoxia and organic matter preservation in sediments as this is much beyond the scope of our paper on AA (lines 543-571).*

Line 651-652: AA are not the main identifiable contributors to all global nitrogen- a qualifier, such as sedimentary or particularly organic N is necessary.

*Reply: Particulate organic N was added.*

Line 660: I believe this is referencing the same paper as the line above (McCarthy 2007), as there is no McCarthy 2006 paper is in references. The citation should be corrected and probably only needs to be cited once at the end of the sentence.

*Reply: These references were corrected and deleted, respectively.*

Line 672: should this say water depth above 200m?

*Reply: 200 m was added.*

Line 698: Can the authors discuss how these calculations compare with past work and direct measurements of %OC and ON which is AAs? For instance, Bronk 2002 suggests dissolved combined AAs only represent ~ 7% of total DON, and Kaiser and Benner 2009 suggest amino acids, amino sugars, and neutral sugars collectively account for 1% to 5% of TOC in the Pacific and Atlantic gyres. Perhaps their values are higher because most of their sampling locations are coastal rather than the open ocean?

*Reply: Lines 113-124 were moved to the Discussion in chapter 4.3 (see lines 677-686) comparing the different AAC/C from the literature to our data. It is striking that the lower AAC/C were obtained under different hydrolysis conditions. However, this needs to be further studied. The above mentioned references were added. In chapter 4.5 on the abundance of i.e. AAC and AAN in the ocean we already compared our data to estimates comparing results of NMR spectroscopy to AA yields by acid hydrolysis and therefore kept the text in this part (lines 783-748):*

"The largest organic carbon pool in the ocean is DOC with an inventory of 632±32 PgC (Carlson and Hansell, 2015; Hansell et al., 2009) and the largest N pool is DON with 77±23 PgN (Gruber, 2008; Bronk, 2002). Dissolved AA are thus the largest AA pool in the ocean even if AA comprise only a minor amount of DOC. We have only few measurements of AA concentrations, which range between 0.1-0.2 mg/L with an average of 0.16 mg/L in all water samples excluding bottom water. Based on these data we can estimate that AA comprise about 200±70 Pg which would contribute about 35±11 Pg AAN and about 89±29 Pg AAC to the oceanic DON and, respectively, DOC pools. **Accordingly, AAC contributes about 14 % to DOC while AAN contributes 45 % to total oceanic DON. This is in the low range of an estimate of 45-86 % AAN based on NMR spectroscopy with acid hydrolysis suggested to recover about half of this AAN pool (Aluwihare et al., 2005).** "

Line 748: I think it would be helpful to mention in paratheses which biogeochemical indicators are not better than POC, as there are others which are not investigated in this study.

*Reply: The indicators not better than POC % used in this work were added.*

**Reply to Reviewer 2:**

Though amino acid composition has been regularly measured as an indicator of organic matter degradation, the varied sources and processes beyond heterotrophic degradation shift the AA composition and limited its application in diverse marine environments. It has been frequently found in recent publication that the DI index was inappropriately applied to samples that were clearly regulated by mechanisms different from Dauwi's dataset. However, the manuscript by Gaye et al. analyzed amino acid composition from amazingly abundant samples covering traps, sediment, suspended particles, the seawater and pore water from varied regions. This provides a complete dataset to reliably reflect the variation in amino acid composition among major pools of amino acids in the ocean. This work well fits the scopes of Biogeoscience. The manuscript was well written and the conclusion is convincing and informative. I would suggest the publication of this work with some minor editions as followings.

*Reply: We thank the reviewer for taking the time to review our paper and for the very encouraging words.*

**Specific comments:**

1. Considering the spatial coverage and large range of concentrations, whether it is suitable to calculate the decay functions (Fig. 3) using the entire dataset?

*Reply: In Figure 3 we excluded Kara Sea samples as resuspended sediments was found to impact SPM and trap samples as we published in Gaye et al. 2007. We added a brief summary on the different areas of sampling to the methods with the references. If we add Kara Sea samples to Fig. 3b the decay function would underestimate organic matter decay and actually confound the decay function as the resuspended sediments are more degraded than SPM and sinking particles from the water column*

2. Were Gluam and Galam measured simultaneously with the amino acids?

*Reply: They were measured simultaneously with amino acids and we clarified this in the methods section of the revised version (lines 213-133). We also added the correction factor of 1.4 for HA which is required when AA and HA are measured with the same hydrolysis.*

3. One thing confused me a lot is where comes the amino acids enriched in pore water, if there is no exchange between pore water and sediment.

*Reply: We decided not to discuss the pore water results any further as this would lengthen the paper and would open up an entirely new discussion on diagenesis in sediments. We are sure that there is exchange between pore water and sediments, which is also shown by further changes of AA spectra in pore water compared with water samples. The increase of non-protein AA in pore water also implies that degradation products are released to the pore water and we added this to the Supplement S3.*

4. Line 268, "carbon and N" to "carbon and nitrogen"

Reply: this was changed

5. Line 406 "Least degraded" to "most degraded"?

Reply: "least degraded" was kept because the DI indeed suggests this because it is not applicable to dissolved AA.

6. Line 436, "increasewith" to "increase with"

Reply: changed

7. Line 725, "Tab. 1" to "Table. 1"

Reply: changed

---

## Author Response (AR2)

Dear Dr. Yuan Shen,

thank you for your valuable comment and the hint at the paper of Peter et al. (2012) with the data to calculate the marine DOM specific degradation index. We have calculated the marine DOM-DI for all water samples and found that it is in a similar range as the DI and that it does not show any trend with water depth. We have included the DOM-DI in the presentation of biogeochemical indicators (lines 281-285), results (lines 356-359; lines 426-428) and discussion (line 674-678) and have included a new supplement S4 which shows that there are no depth dependent trends in water and pore water samples. This furthermore supports our statement that total dissolved amino acids show no depth dependent changes in AA composition (line 655) in our data set and with the hydrolysis conditions we used.

Best regards,

Birgit Gaye